Qi *et al.* Genome Biology     (2025) 26:180

**RESEARCH**

# Long-term DNA methylation changes mediate heterologous cytokine responses after BCG vaccination

Cancan Qi[1,2,3†], Zhaoli Liu[1,2†], Gizem Kilic[4], Andrei S. Sarlea[4], Priya A. Debisarun[4], Xuan Liu[1,2], Yonatan Ayalew Mekonnen[1,2,5], Wenchao Li[1,2], Martin Grasshoff[1,2], Ahmed Alaswad[1,2], Apostolos Petkoglou[1,2], Valerie A. C. M. Koeken[1,2,4,6], Simone J. C. F. M. Moorlag[4], L. Charlotte J. de Bree[4], Vera P. Mourits[4], Leo A. B. Joosten[4,7], Yang Li[1,2,4†], Mihai G. Netea[4,8†] and Cheng-Jian Xu[1,2,4*†]

†Cancan Qi and Zhaoli Liu contributed equally and share the first authorship.

†Yang Li, Mihai G. Netea and Cheng-Jian Xu contributed equally and share the last authorship.

*Correspondence:
Xu.Chengjian@mh-hannover.de

¹ Centre for Individualised Infection Medicine (CiiM), a joint venture between the, Helmholtz Centre for Infection Research (HZI) and Hannover Medical School (MHH), Hannover, Germany
Full list of author information is available at the end of the article

## Abstract

**Background:** Epigenetic reprogramming shapes immune memory in both innate (trained immunity) and adaptive immune cells following Bacillus Calmette-Guérin (BCG) vaccination. However, the role of dynamic DNA methylation changes in post-vaccination immune responses remains unclear.

**Results:** We established a cohort of 284 healthy Dutch individuals, profiling genome-wide DNA methylation and cytokine responses to ex vivo stimulation at baseline, 14 days, and 90 days post-BCG vaccination. We identified distinct patterns of DNA methylation alternations in the short- and long-term following BCG vaccination. Moreover, we established that baseline DNA methylation profiles exert influence on the change in interferon-γ (IFN-γ) production upon heterologous (*Staphylococcus aureus*) stimulation before and after BCG vaccination. Specifically, we identified the regulation of kisspeptin as a novel pathway implicated in the modulation of IFN-γ production, and this finding has been substantiated through experimental validation. We also observed associations between BCG-induced DNA methylation changes and increased IFN-γ and interleukin-1 β (IL-1β) production upon *S. aureus* stimulation. Interestingly, by integrating with genetic, epigenetic, and cytokine response data from the same individuals, mediation analysis demonstrated that most of the identified DNA methylation changes played a mediating role between genetic variants and cytokine responses; for example, the changes of cg21375332 near *SLC12A3* gene mediated the regulation of genetic variants on IFN-γ changes after BCG vaccination. Sex-specific effects were observed in DNA methylation and cytokine responses, highlighting the importance of considering sex in immune studies.

**Conclusions:** These findings provide deeper insights into immune response mechanisms, crucial for developing effective epigenetic-based medical interventions for personalized medicine.

**Keywords:** BCG vaccination, Cytokines response, Trained immunity, DNA methylation, Systems biology

## Background

Host immune responses are traditionally classified into innate and adaptive, with only the latter initially thought to have the ability to develop immunological memory. However, in recent years, a growing body of evidence has shown that innate immunity can also exhibit memory characteristics [1]. Studies have shown explicitly that Bacillus Calmette-Guérin (BCG) vaccination can induce innate immune memory through epigenetic reprogramming of myeloid cells and their bone marrow progenitors [2–6]. This non-specific innate immune memory can provide cross-protection against unrelated pathogen and is referred to as "trained immunity" [7, 8]. In addition, BCG vaccination induces classical adaptive immune memory, with T-cell-derived IFN-γ responses being relevant for protection against tuberculosis [2].

A distinguishing feature of the trained innate immune cells is their ability to mount a stronger transcriptional response compared to naïve cells when challenged with a pathogen. Recent research has identified various factors that can influence the induction of trained immunity, such as circulating inflammatory proteins [9], metabolites [10], and transcriptomic profiles of immune cells [11–13], and the gut microbiome [14]. Studies have shown that after being vaccinated with BCG, systemic inflammation is reduced at both the protein and transcription levels [9, 11, 12, 15]. Additionally, the production of cytokine following BCG vaccination is associated with the abundance of microbial genomes, which in turn influences circulating metabolites [14].

One of the key molecular mechanisms involved in the induction of trained immunity is the epigenetic reprogramming of immune cells. Chromatin accessibility and histone marks have been the mechanisms most studied in this regard [7, 16]. Another important mechanism for epigenetic gene transcription regulation is through modulation of DNA methylation. Previous studies have demonstrated the association of DNA methylation with infection and immune memory [17, 18], showing how infection-induced changes in DNA methylation regulate the transcriptional response to infection and contribute to short-term memory in innate immune cells [17]. Demethylation of enhancer elements mediated by transcription factors (TFs) binding may allow for a faster response to a secondary infection and thus plays a crucial role in innate immune memory [19]. More recently, studies have shown that BCG vaccination leads to changes in DNA methylation [20] in monocytes and T cells of children [21]. We have previously suggested that DNA methylation changes after BCG vaccination in adults as well [22], but a comprehensive analysis of the impact of DNA methylation on innate immune memory is missing. To better understand the molecular basis of trained immunity, we investigated whether DNA methylation plays an important role in this process.

In this study, we hypothesize that in vivo BCG vaccination induces changes in DNA methylation associated with a trained immunity response. To test this hypothesis, we longitudinally assessed ex vivo cytokine production upon *Staphylococcus aureus* stimulation, and genome-wide DNA methylation at baseline, 14 days, and 90 days after BCG vaccination in a cohort of 303 healthy volunteers from the Human Functional Genomics Project (300BCG cohort, www.humanfunctionalgenomics.org). Using integrative system biological approaches, we assessed the dynamic changes in DNA methylation following BCG vaccination and their associations with BCG-induced immunological memory, focusing on ex vivo cytokine production changes of four cytokines upon *S. aureus*

stimulation. Specifically, monocyte-derived cytokines include tumor necrosis factor-alpha (TNF-α), interleukin-1β (IL-1β), and interleukin-6 (IL-6), while T cell-derived cytokines are represented by interferon-gamma (IFN-γ). We further validated our findings in an independent cohort (BCG booster trial [23]) and a number of the identified pathways in in vitro experiments.

## Results

### Study design and global DNA methylation variability

This study was conducted in the 300BCG cohort, and the details of the cohort have been previously reported [9]. In brief, 303 healthy volunteers received BCG vaccination and whole blood samples were collected before (T0), 14 days (T14), and 3 months (T90) after vaccination. The genome-wide DNA methylation profiles were then measured on high-quality DNA isolated from the whole blood using Illumina Epic arrays. Additionally, the production of cytokines (IL-1β, IL-6, TNF-α, IFN-γ) from peripheral blood mononuclear cells (PBMCs) was measured after stimulation with *Staphylococcus aureus* at T0 and T90. The fold change (FC) of cytokine production between T90 and T0, referred to as trained immunity markers in this study, was used as a measure of BCG-induced trained immunity response. Plasma inflammatory protein concentrations were also obtained at three time points (Fig. 1A). After rigorous quality control, downstream analyses included 284 subjects, consisting of 126 males and 158 females. The mean age was 25 years (standard deviation 10.55, ranging from 18 to 71), with 86.3% of participants under 30 years old (Additional file 1: Fig. S1), and the mean BMI was 22.48 kg/m$^2$ (Additional file 2: Table S1). Epigenome-wide association studies (EWAS) were performed on 751,564 high-quality probes from the three different time points of these 284 subjects. The EWAS aimed to examine (1) the epigenetic modifications induced by BCG vaccination and (2) the association between DNA methylation and changes of ex vivo cytokine production capacity induced by vaccination (Fig. 1B).

The univariate associations between the principal components (PCs) calculated from the whole blood DNA methylation data at each time point, covariates, and estimated cell proportions [24] are depicted in Fig. 1C and Fig. S1 (Additional file 1). The first 30 PCs accounted for 35% of the total variation in the blood methylome at T0. The strongest associations were observed between the cell proportions, particularly neutrophils, and the top two DNA methylation PCs. The first DNA methylation PC was also associated with batch effects (sample plate), while age, sex, smoking behavior, and body mass index (BMI) showed no significant correlations with the top two PCs.

### BCG vaccination induced changes in DNA methylation

The changes in DNA methylation over time after BCG vaccination were assessed using a mixed effects model that accounted for age, sex, batch effects (sample plate) and estimated cell proportions as covariates, with sample ID as a random effect. In total, we identified 11 CpG sites that were significantly (false discovery rate (FDR) < 0.05) changed following BCG vaccination (Fig. 2A, Additional file 2: Table S2). Among these, five CpG sites showed increased methylation, while six CpG sites displayed decreased methylation 90 days after vaccination. Notably, the changes observed at T14–T0 for eight out of the 11 CpG sites were converse to those observed

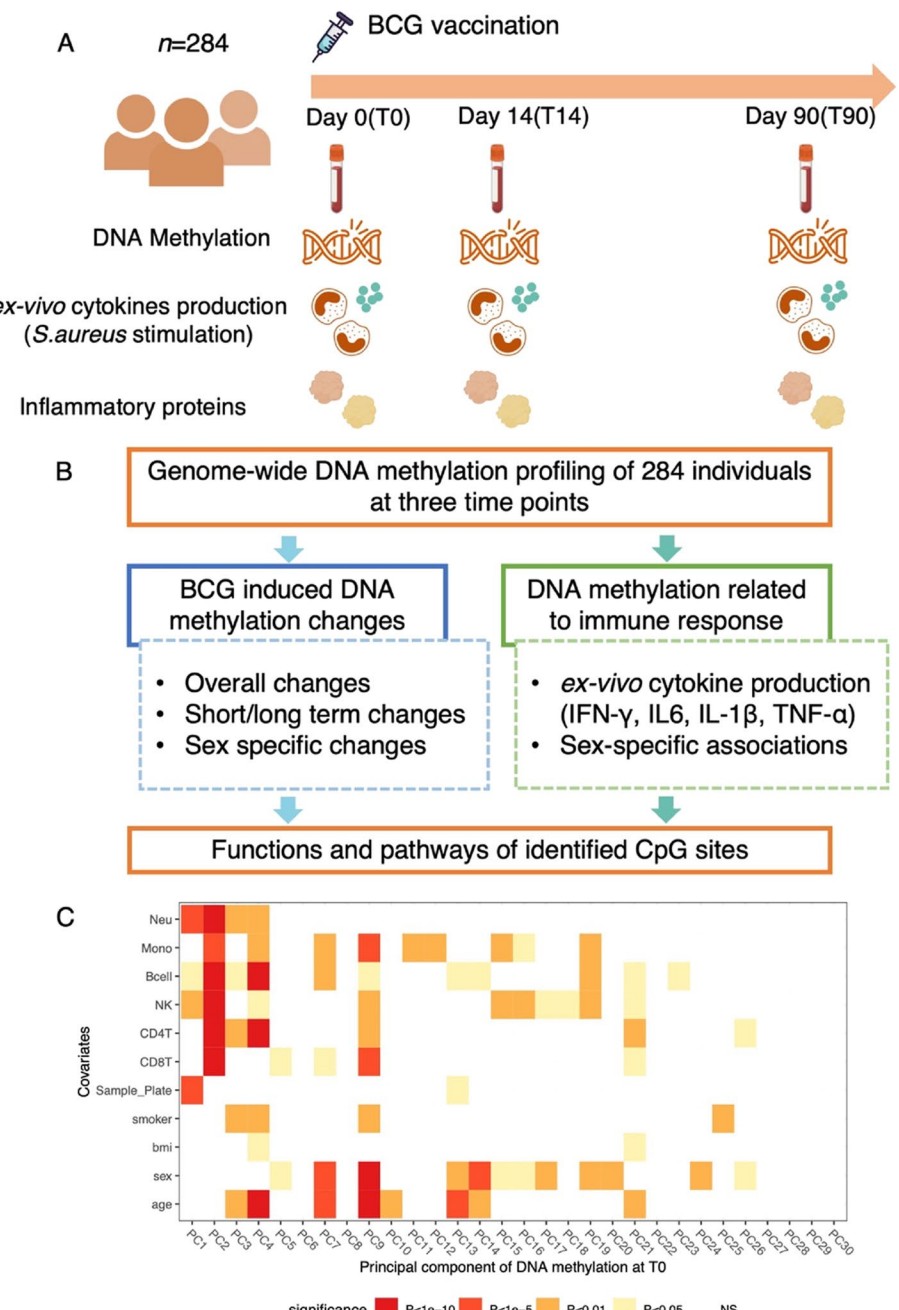

**Fig. 1** Study design and data exploration. **A** Overview of the study design. **B** Main analysis in this study. **C** Heatmap of association between covariates, estimated cell proportions, and DNA methylation at T0, which are represented as the top 30 PCs, capturing 35% of the variance. The association was performed with a univariable linear regression model; different colors in the figure indicate different levels of significance. BCG, Bacillus Calmette–Guérin; PC, principal component

at T90–T0 (Fig. 2B). This observation potentially indicates the presence of distinct epigenetic mechanisms during different time intervals after BCG vaccination. To verify this finding, we further checked the 73 CpG sites with suggestive significance ($P < 1 \times 10^{-5}$), and identified consistent patterns (Fig. 2C). Pathway analysis of the genes

annotated to the 11 CpG sites revealed enrichment in pathways related to Human papillomavirus infections, VEGFA-VEGFR2 signaling, and tight junction, which were reported to be related to the immune response to infections previously [25, 26] (Additional file 1: Fig.S2 A).

We next checked the association between the identified CpG sites and other traits using the EWAS catalog (http://www.ewascatalog.org) (Additional file 2: Table S3). Five out of the 11 CpG sites have been previously associated with age. Two of these sites were reported to be associated with proteins that are involved in B cell development. For instance, cg06221058 (mapped to gene *MYH9*) was reported to be associated with age and IGMH (Immunoglobulin heavy constant Mu) protein levels. cg06013215 (mapped to gene *PTGER4*) was reported to be related to protein amounts of IGLL1 (Immunoglobulin lambda-like polypeptide 1) and FCER2 (Fc Epsilon Receptor II).

Considering the cell-type specificity of DNA methylation markers, the methylation level of these 11 significant CpG sites was then correlated with the estimated cell proportions at each time point (Additional file 1: Fig. S2B). At all three time points, strong correlations were observed between the CpG sites and estimated neutrophil proportion, with T90 showing the strongest associations. Consistent patterns were identified when correlating the CpG sites with cell counts of neutrophil subtypes measured by flow cytometry (Additional file 1: Fig. S2 C). The top correlated neutrophil subtypes included CD10 +CD66b, CD10 +CD62L, and PDL1-CD62LP neutrophils. The long-term functional reprogramming of neutrophils has been previously reported [6], and our findings provide additional evidence of BCG-induced DNA methylation changes associated with neutrophils.

### BCG vaccination induced short-term and long-term changes in DNA methylation

As shown in Fig. 2B–C, the observed patterns of DNA methylation changes suggested the involvement of distinct epigenetic mechanisms in the short- and long-term effects

(See figure on next page.)

**Fig. 2** BCG vaccine induced short-term and long-term methylation changes. **A** Manhattan plot showing DNA methylation changes over time after BCG vaccination. Differentially methylated CpG sites ($N = 11$) with false discovery rate (FDR) < 0.05 were highlighted and labeled with the probeID. **B** Heatmap of hierarchical clustering on changes of identified 11 CpG sites at T14 and T90 compared to T0, which are represented as the differences in DNA methylation beta values. **C** Bar plot of the number of CpG sites with significant change over time with $P < 1 \times 10^{-5}$ in different groups. Down_down group indicates DNA methylation was decreased at both T14 and T90 compared with T0; similarly, down_up, decreased at T14 and increased at T90, up_down, increased at T14 and decreased at T90, and up_up, increased at both T14 and T90. **D–F** Change patterns and pathway enrichment analysis of CpG sites identified which assessed the DNA methylation changes at day 14 and day 0 (T14–T0, **D**), day 90 and day 0 (T90–T0, **E**), as well as day 90 and day 14 (T90–T14, **F**) with $P < 1 \times 10^{-5}$. Each panel includes line charts showing the patterns of changes with two plots showing decrease (upper) and increase (lower) change, respectively, and the top enriched pathways and chromatin state were labeled right to each line chart. The pathway enrichment analyses were performed by CPDB (http://cpdb.molgen.mpg.de/) an online tool of gene set analysis; the chromatin enrichment analysis was performed by eFORGE (https://eforge.altiusinstitute.org/). **G** Bar plot showing the number of decreased and increased CpG sites identified from T14–T0, T90–T0, and T90–T14, respectively. Blue represents hypomethylated sites and yellow represents hypermethylated sites. The *P* value on the top of the figure shows the enrichment of hypomethylated/hypermethylated sites in increase/decrease status (Fisher's exact test)

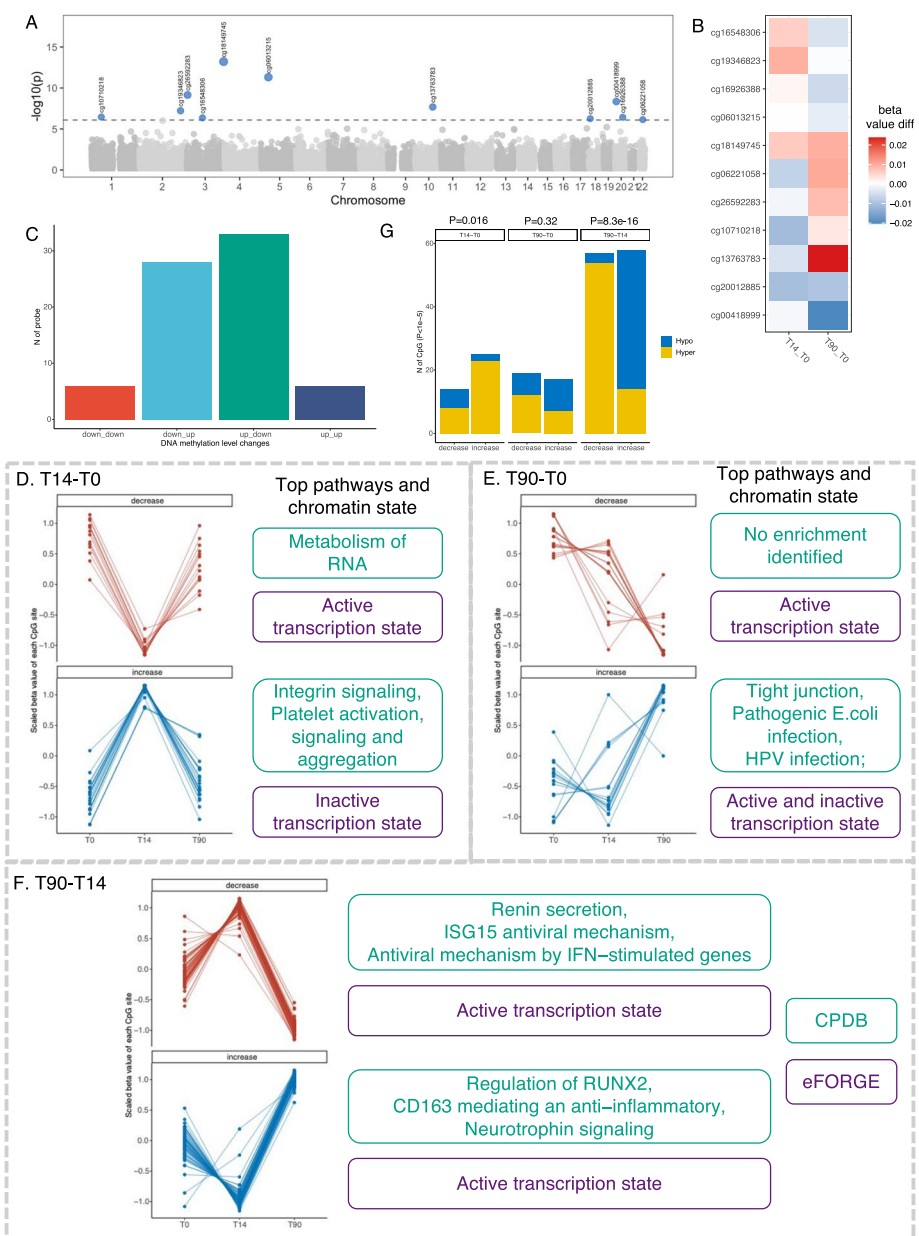

**Fig. 2** (See legend on previous page.)

following BCG vaccination. To gain further insights into these effects, we compared the DNA methylation changes between day14 and day0 (T14–T0), day90 and day0 (T90–T0), as well as day90 and day14 (T90–T14) respectively, using mixed effects models. Three CpG sites from the T14–T0 model, five from the T90–T0 model, and sixty-four from the T90–T14 model were identified with genome-wide significant changes (FDR < 0.05). In order to capture the overall dynamics of DNA methylation after BCG vaccination, CpG sites reaching suggestive significance threshold ($P < 1 \times 10^{-5}$) were also included in the downstream exploratory analysis (Additional file 2: Table S4–S6).

In the T14–T0 model, which represents the short-term effect of BCG vaccination on methylome, we identified 39 CpG sites ($P < 1 \times 10^{-5}$). Among these, 14 CpG sites exhibited decreased methylation at T14 compared to T0, while 25 CpG sites showed increased methylation. Notably, all of these sites demonstrated an opposite direction of change when comparing T90–T14 (Fig. 2D, Additional file 1: Fig. S3 A, D). This suggests that the later time period, from T14 to T90, might represent a recovery phase following BCG vaccination. Using functional enrichment tools experimentally derived Functional element Overlap analysis of ReGions from EWAS(eFORGE) [27], we found that the demethylated CpG sites were enriched in active transcription states, and the genes mapped to these sites were enriched in pathways involved in the metabolism of RNA, indicating an active transcriptional activity of the demethylated sites at T14. In contrast, the methylated (the increase in methylation) CpG sites were enriched in an inactive transcription state, and were involved in pathways related to integrin signaling and platelet activation and aggregation (Fig. 2D, Additional file 1: Fig. S4D). These results suggested that the short-term DNA methylation alterations induced by BCG vaccination may be involved in the regulation of transcription activity and may return to their original levels over time.

The long-term effect model, T90–T0, revealed 17 methylated CpG sites and 19 demethylated CpG sites ($P < 1 \times 10^{-5}$, Fig. 2E, Additional file 1: Fig. S3B, D) associated with BCG vaccine. These methylated CpG sites exhibited enrichment in both active and inactive transcription states, and the genes mapped to these CpG sites were enriched in pathways such as tight junction and bacterial and viral infection (Fig. 2E, Additional file 1: Fig. S4E). This suggests that the long-term effect of BCG on epigenetics may be related to the response to pathogenic infection. The long-term effect of BCG vaccination on epigenetics at 90 days post-vaccination reflects a lasting epigenetic memory. Interestingly, the majority of the increased CpG sites (64.7%) first slightly decreased at T14 and then increased at T90, indicating that the majority of the increased changes at T90 actually occurred after T14 (Fig. 2E, Additional file 1: Fig. S3B, D). These results suggested a "late" epigenetic effect of vaccination, in contrast to the "immediate" transcriptional effect of vaccination [13].

To gain a deeper understanding of the "late" effect of BCG vaccination, we conducted a comparison of methylation profiles between T90 and T14. Through this analysis, we identified 115 CpG sites that displayed suggestively significant changes ($P < 1 \times 10^{-5}$). Among these, 58 CpG sites showed increased methylation at T90 compared to T14 after initially being demethylated at T14, surpassing even the levels observed at T0. In contrast, the demethylated CpG sites exhibited the opposite pattern of methylation changes (Fig. 2F, Additional file 1: Fig. S3 C, D). The genes mapped to the identified CpG sites were enriched in pathways related to active transcription state, inflammatory response (e.g., ISG15 antiviral mechanism, antiviral mechanism by IFN-stimulated genes, and CD163 mediated anti-inflammatory) and nervous system (e.g., renin secretion and neurotrophin signaling) (Fig. 2F, Additional file 1: Fig. S4 F). Interestingly, we found that the methylated CpG sites identified from the comparison of T90 vs T14 were enriched in hypo-methylated sites, while the demethylated CpG sites were enriched in hypermethylated sites (Fisher's exact test, $P = 8.31 \times 10^{-6}$, Fig. 2G, Additional file 1: Fig. S4 C). However, the increased CpG sites identified from the T14 vs T0 comparison model were

enriched in hyper-methylated sites, contrasting with the pattern observed in the comparison of T90 vs T14 (Additional file 1: Fig. S4 A–B).

We conducted a replication study to validate our findings using an independent cohort. In this study, 17 individuals were vaccinated with BCG, and blood samples were collected before vaccination (T0) and 3 months (T90) after vaccination, followed by DNA methylation profiling. The assessment of ex vivo cytokine production followed the same protocol as that of the discovery cohort. We compared the difference in DNA methylation levels of 36 CpG sites ($P < 1 \times 10^{-5}$ in the discovery cohort) between T90 and T0 in the replication cohort. Among these, 21 CpG sites showed the same direction, and two out of the 21 CpG sites showed nominal significance ($P < 0.05$) (Additional file 2: Table S7).

To further investigate the effect of other potential confounders that may influence the results, several sensitivity analyses were performed. In the sensitivity analyses incorporated 12 cell types estimated by EpiDISH [28], a total of 263 CpG sites were tested ($P < 1 \times 10^{-5}$, including T0–T14–T90, T14–T0, T90–T0, and T90–T14). Among these, 214 CpG sites (81.4%) remained suggestively significant ($P < 1 \times 10^{-5}$) after adjusting for the 12 cell types (Additional file 2: Table S8). Additionally, we assessed the impact of seasonality on the BCG-associated CpG sites. The results showed that 71.5% of the top CpG sites remained suggestively significant ($P < 1 \times 10^{-5}$) after seasonality adjustment (Additional file 2: Table S8), and 95.4% of these BCG-associated CpG sites are not associated with seasonality ($P > 0.05$, Additional file 2: Table S9). Given that the study population primarily consisted of young adults under 30 years old, we further performed sensitivity analyses in the subgroup of participants older than 30. In this subgroup, 46% of CpG sites reached nominal significance in the association with BCG effect (Additional file 2: Table S8). These findings suggest that the results of this study are most applicable to young adults. Further research in older populations is warranted to better understand these effects.

### BCG-induced DNA methylation changes were associated with the alteration of plasma inflammatory proteins

We previously reported that BCG vaccination reduced circulating inflammatory markers at both 14 days and 90 days after vaccination [9]. Given the significant changes in DNA methylation profiles upon BCG vaccination observed in this study and that the enriched pathways were related to immune responses to infections, we further investigated whether BCG-induced DNA methylation changes were associated with the altered concentration of circulating inflammatory proteins. In our group's previous findings, we identified 24 proteins exhibiting significant short-term decreases and 10 proteins that exhibited long-term decreases after BCG vaccination [9]. We correlated these proteins with CpG sites identified in T14 vs T0 (No. CpG = 39) and T90 vs T0 (No. CpG = 36) comparison, respectively. We found that the DNA methylation changes in four CpG sites identified from T14 vs T0 comparison were significantly associated with alterations in two proteins upon BCG vaccination, with FDR < 0.05 (Additional file 2: Table S10). Specifically, two CpG sites were associated with CD6 (cg26624673 near gene *ADHFE1* and *MYBL1*, and cg10813029 near gene *TGIF1* and *DLGAP1*), and two were associated with OPG (cg20782252 near gene *MAD1L1* and *ELFN1*, and cg12598528 near gene *KCNT2*).

In addition, we identified 77 CpG-protein pairs that reached nominal significance ($P <$ 0.05), among which we observed a consistent pattern in which the CpG sites exhibiting increased methylation at T14 in comparison to T0 were inversely correlated with proteins that exhibited a decrease at T14 relative to T0 (Additional file 1: Fig. S5 A). However, the CpG sites identified from the comparison of T90 vs T0 did not show any significant association with protein abundance (FDR < 0.05, Additional file 2: Table S10, Additional file 1: Fig. S5B). These results indicate that BCG-induced DNA methylation changes are associated with reduced concentrations of inflammatory proteins in the circulation, particularly regarding DNA methylation changes from T0 to T14. This suggests that short-term DNA methylation changes may play a role in the reduction of circulating inflammatory proteins induced by BCG vaccination.

### Baseline epigenetic profiles were associated with the trained immunity responses after BCG vaccination

Next, to further understand whether differences in baseline DNA methylation influence BCG-induced trained immunity (TI), we assessed the association between baseline DNA methylation status and BCG-induced TI responses. TI responses were defined as the fold changes of cytokine productions stimulated with S. aureus at T90 compared to T0 (Fig. 1). We measured both the monocyte-derived cytokines (IL-1β, IL-6, and TNF-α), and IFN-γ, which primarily represents T-cell-derived immune response. Our results revealed that 41 CpG sites at baseline were significantly associated with the increased IFN-γ production capacity at T90 compared to T0, denoted as TI (IFN-γ) (FDR < 0.05, Fig. 3A, Additional file 2: Table S11–S12). The strongest association was observed at cg16685860 ($P = 6.02 \times 10^{-10}$), located near gene *PLD2*. We identified 22 out of 41 CpG sites that were positively associated with TI (IFN-γ). The genes mapped to these sites were enriched in pathways including phospholipases, kisspeptin receptor system, and LPA receptor mediated events. Conversely, 19 out of the 41 sites were negatively associated with TI (IFN-γ), and the genes mapped to these sites were enriched in the apoptosis pathway and FAS pathway, which was also related to programmed cell death (Additional file 1: Fig. S6 A). We did not find any significant associations between baseline DNA methylation and TI (IL-1β) or TI (TNF-α), except for one CpG site (cg07586956, mapped to *PCBP1* and *C2orf42* genes) that showed an association with TI (IL-6) ($P = 5.40 \times 10^{-10}$). Furthermore, we assessed the association between DNA methylation at 14 and 90 days after BCG vaccination and TI (Additional file 2: Table S13–S14). We found a total of 9 CpG sites at T14 and 95 CpG sites at T90 were linked to TI (IFN-γ) (Additional file 2: Table S13). For TI (IL-1β), we identified 32 CpG sites at T14 and only three CpG sites at T90 (Additional file 2: Table S14), showing different patterns compared to the analysis of TI (IFN-γ), where more sites were identified at T90.

We further checked the association of the identified CpG sites with other traits by utilizing the EWAS catalog (Additional file 2: Table S12). Among the 41 CpG sites, 18 were previously found to be associated with age (Additional file 1: Fig. S6B). Interestingly, we discovered that five CpG sites (cg03433260 near gene *BID*, cg09391860 near gene *ZNF335* and *MMP9*, cg18591181 near gene *NIM1 K*, cg25937862 near gene *MRC2*, and cg22032521 near gene *HYLS1*) showed associations with five protein levels (EBAG9, GZMK, HAPLN4, LYN, and OSM) that have been previously reported

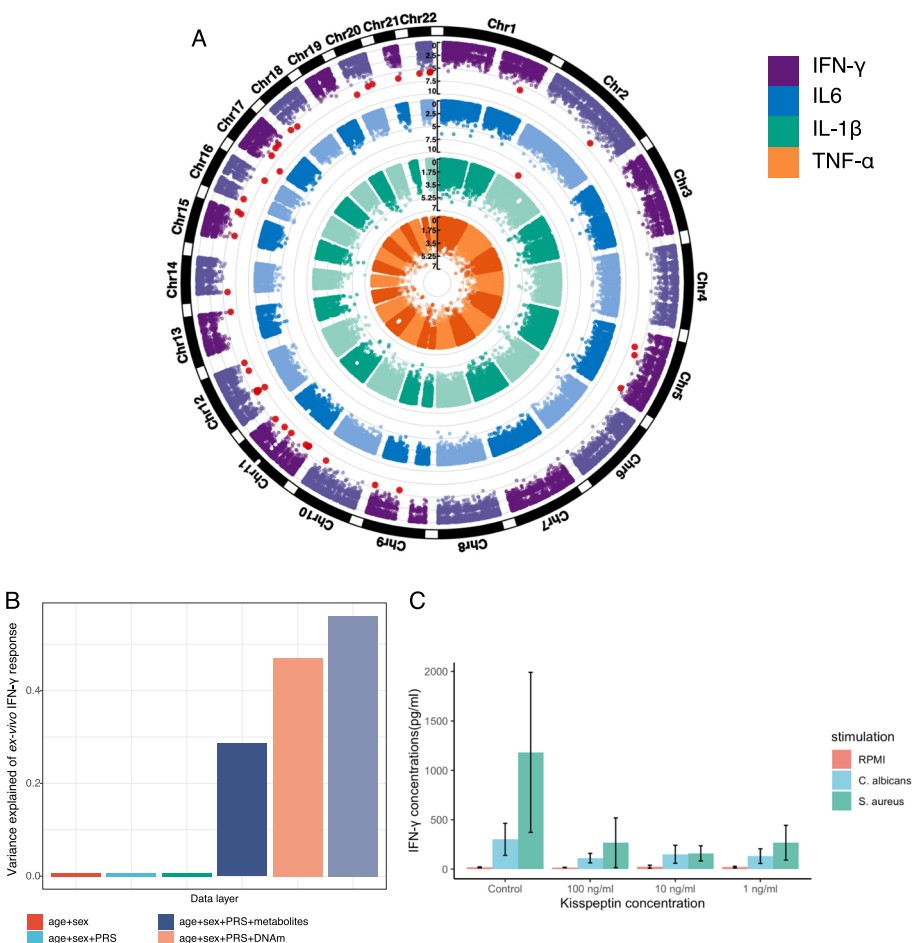

**Fig. 3** Baseline epigenetic markers and ex vivo cytokine production changes. **A** Manhattan plot showing the association of baseline DNA methylation level and ex vivo immune response, represented by fold-change of TNF-α, IL-1β, IL6, and IFN-γ production stimulated by *S. aureus* 90 days after BCG vaccination compared to baseline (from middle to outside). Genome-wide significant CpG sites (FDR < 0.05) are highlighted in red. **B** Bar plot showing the variation in ex vivo IFN-γ response explained by data from various omics layers, including general information (age and sex), genetics (PRS), baseline DNA methylation, baseline inflammatory proteins, and baseline plasma metabolites. **C** Experimental validation. Chr, chromosome; AUC, area under the curve; CI, confidence interval; PRS. polygenic risk score; DNA methylation: DNA methylation; protein: inflammatory proteins

in blood. We associated INF-γ related CpG sites with cell counts measured by flow cytometry, and identified that T cell numbers (both CD4 + and CD8 +) were associated with the greatest number of CpG sites (Additional file 1: Fig. S6 C).

Then, we investigated how much variance in TI (IFN-γ) could be explained by different omics layers, including polygenetic risk score (PRS) [29], inflammatory proteins, metabolites, and DNA methylation. Comparing age, sex, and PRS and baseline inflammatory proteins, we discovered that baseline DNA methylation explained the largest proportion of variance of TI (IFN-γ), followed by the baseline metabolites (Fig. 3B). A combined model of all data layers explained 56.1% variance for TI (IFN-γ). Further randomization tests confirm the significance of the finding (*P* value < 0.001) (Additional file 1: Fig. S6D). We also checked if SNPs located proximal to the

CpG sites associated with TI (IFN-γ) were enriched in cytokine-QTL of IFN-γ, but we did not identify any significant enrichment (Additional file 2: Table S15, Additional file 1: Fig. S6E).

Next, we validated the associations between baseline DNA methylation and TI (IFN-γ) in the replication cohort, using the same model as in the discovery cohort. Out of 40 CpG sites, we identified five CpG sites that could be replicated with nominal significance ($P < 0.05$) and consistent direction of effect (Additional file 2: Table S16).

Sensitivity analyses were performed to investigate the effect of other potential confounders on the association between baseline DNA methylation and cytokine production. In the sensitivity analyses, where 12 cell types were included as covariates in the model, all the results showed a consistent direction of effect with the original analysis, and 39% of CpG sites remained suggestive significant ($P < 1 \times 10^{-5}$) (Additional file 2: Table S17). This indicates that some of the CpG sites might be driven by specific immune cell types that should be interpreted with caution. Additionally, sensitivity analyses revealed that these CpG sites associated with IFN-γ or IL-6 production were not associated with seasonality ($P > 0.05$, Additional file 2: Table S18).

### Kisspeptin modulates IFNγ production capacity

According to the pathway enrichment analysis of IFN-γ associated CpG sites, the kisspeptin receptor system pathway showed the strongest enrichment, with the lowest $P$ value and the largest number of enriched genes (Additional file 1: Fig. S6 A). Kisspeptins are proteins encoded by the *KISS1* gene that have been initially described to inhibit metastases in cancer and induce secretion of gonadotropin-releasing hormone (GnRH), but recently, an increasing role in immunomodulatory effects has been described [30]. To further understand whether kisspeptins modulate IFN-γ production, we performed a functional validation experiment. Human peripheral blood mononuclear cells were incubated for 1 h with various concentrations of recombinant kisspeptin-10 (from 1 ng/ml to 100 ng/ml), followed by stimulation with either heat-killed *C. albicans* or *S. aureus* for 7 days. Kisspeptin significantly inhibited IFN-γ production capacity, validating its immunomodulatory role (Fig. 3C). In contrast, no effects on IL-17 and IL-22 production, another two cytokines produced by T cells, were observed (not shown). This demonstrates the inhibitory effect of kisspeptin on IFN-γ production and argues that DNA methylation of *KISS1* likely impacts interferon production.

### BCG-induced DNA methylation changes were associated with ex vivo cytokine production changes

In order to investigate if the BCG-induced DNA methylation changes are associated with cytokine production changes, we further performed association analyses between Trained immunity and the differences of DNA methylation levels (DNAm-C) between any two time points (T14–T0, T90–T0, and T90–T14, Additional file 2: Table S19). Notably, T90–T0 DNAm-C showed a significant association with trained immunity for all four traits, with the following numbers of significant associations: TNF $N = 2$, IFN-γ $N = 28$, IL-1β $N = 14$, and IL-6 $N = 3$ (FDR < 0.05, Additional file 1: Fig. S7 A). This indicates that BCG-induced long-term DNA methylation changes were associated with ex vivo cytokine production changes. To better understand the roles of the trained

immunity related DNAm-C of T90–T0, we performed the pathway enrichment analyses of the genes mapped to suggestively significant CpG sites with a *P* value lower than $1 \times 10^{-5}$ (Additional file 1: Fig. S8). The trained immunity response, as assessed by IL-1β production in response to a heterologous stimulus from the T90–T0 model, was associated with enrichment in changes in DNA methylation for genes involved in Kinesins and Golgi-to-ER retrograde traffic. Kinesins are motor proteins that ensure the transport of cellular cargo, with an emerging role in supporting immune processes as well [31, 32]. On the other hand, changes in IFN-γ heterologous production were enriched in changes in DNA methylation of genes involved in the mTOR signaling pathway (a well-known pathway in T-cell activation and trained immunity [33, 34]), the VEGFA-VEFGR2 signaling pathway, and other immune-related pathways (e.g., IL-2 family signaling and IL-18 signaling pathways). These findings provide a further understanding of the functional implications of the DNAm-C in relation to cytokine production changes.

Interestingly, we observed that some of the CpG sites from the association model between trained immunity and DNAm-C (T90–T0) (TNF-α $N = 11$, IFN-γ $N = 7$, IL-1β $N = 7$, and IL-6 $N = 5$, Additional file 1: Fig. S7 C) exhibited nominal significant changes upon BCG vaccination (*P* value <0.05). However, the CpG sites from the association analyses between trained immunity and DNAm-C (T14–T0) or (T90–T14) did not show significant changes upon BCG vaccination. These findings suggest that the BCG-induced long-term DNA methylation changes, rather than those observed during the vaccine phase (T14–T0) or recovery phase (T90–T14) changes, may play a more important role in the trained immunity response. Furthermore, it indicates that the development of immune-related epigenetic memory relies on long-term epigenetic changes. We also observed that TI (IFN-γ)-C was associated with both baseline DNA methylation and DNAm-C of T90–T0 with a higher number of identified CpG sites compared to other cytokines. This finding is consistent with previous findings that reported enrichment of genes near CpG sites with long-term BCG effect in IFN-related pathways [20].

### Bidirectional mediation between DNA methylation changes and ex vivo cytokine production changes

The underlying causality of the significant association between trained immunity and DNAm-C remains unknown. To investigate this, we utilized the genotype data from 300BCG to infer a potential in silico causal relationship between trained immunity and DNA methylation changes (T90–T0) through mediation analysis [35]. We initially identified SNP—trained immunity—DNAm-C groups that exhibited significant association between every two variables and subsequently conducted bidirectional mediation analysis. In the mediation analysis of Direction1, we hypothesized the effects of SNPs on trained immunity were mediated by DNAm-C, and treated DNAm-C as mediator and trained immunity as outcomes. In the analysis of Direction2, trained immunity was treated as the mediator and DNAm-C as outcome (Fig. 4A). The majority of significant mediation results were identified for TI (IFN-γ). Specifically, seven CpG sites showed unidirectional mediation effects in Direction1 ($P_{\text{Direction1}} < 0.05$ and $P_{\text{Direction2}} > 0.05$), two CpG sites of unidirectional mediation effects in Direction2 ($P_{\text{Direction2}} < 0.05$ and $P_{\text{Direction1}} > 0.05$), and six CpG sites with bidirectional mediation effects ($P_{\text{mediation}} < 0.05$, Fig. 4B, Additional file 2: Table S20). In addition, we also identified two CpG sites that

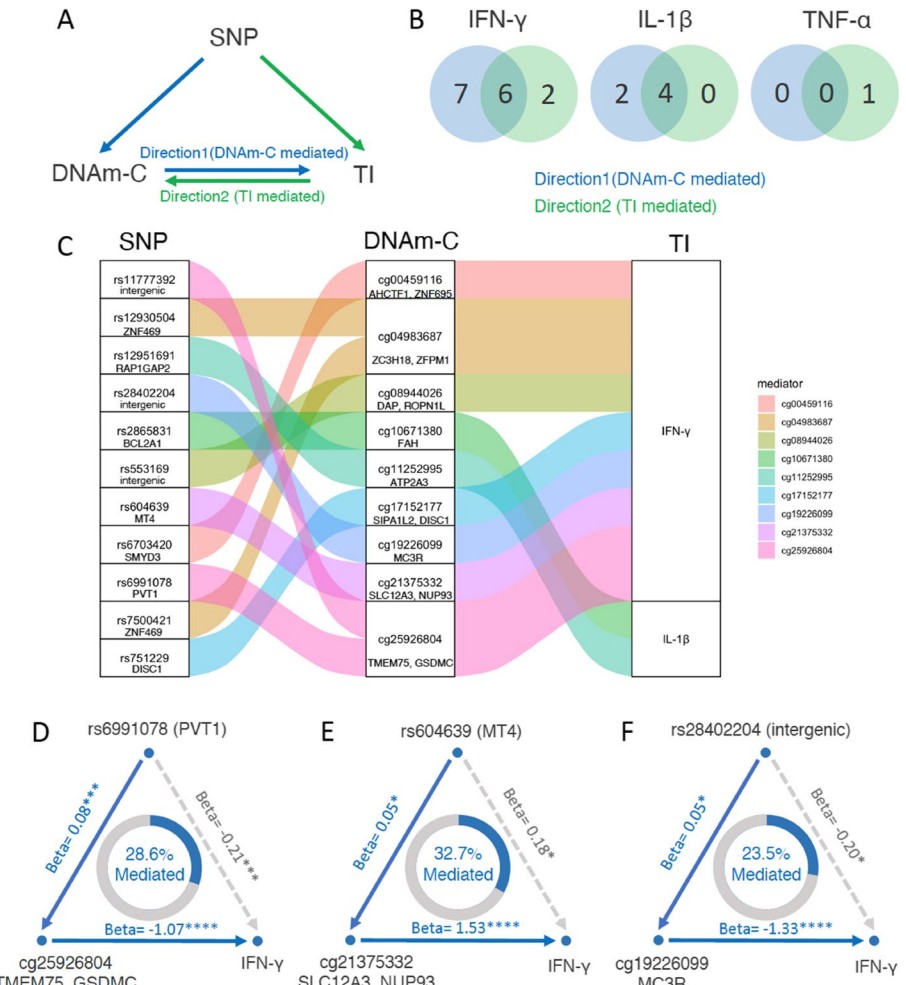

**Fig. 4** Causal relationship inference by bidirectional mediation analysis. **A** Framework of bidirectional mediation analysis between SNPs, DNA methylation changes (DNAm-C) and cytokine production changes (TI). **B** Number of CpG sites that were significant in mediation results of Direction1 (from DNAm-C to TI), Direction2 (from TI to DNAm-C) and both, for IFN-γ (left) and IL-1β (right). **C** Sankey diagram showing the inferred causal relationship network of Direction1 with mediation *P* value < 0.05. **D**–**F** Example of causal relationships between SNP, DNAm-C, and TI inferred by bidirectional mediation analysis. The beta coefficient and significance are labeled at each edge and the proportions of mediation effect are labeled at the center of ring charts. See also Table S14

mediated the effect of genetic variants on IL-1β changes (Direction1), along with four with bidirectional mediation effects, but none of the CpG sites were significant in Direction2 (Fig. 4B, Additional file 2: Table S20).

This approach revealed a total of 18 unidirectional mediation linkages: Direction1 analysis (DNAm-C mediated) identified 11 significant linkages involving 9 CpG sites and two cytokines (IFN-γ and IL-1β) (Fig. 4C); on the other hand, Direction2 analysis (trained immunity mediated) revealed 7 significant linkages consisting of three CpG sites and two cytokines (IFN-γ and TNF-α). More linkages and CpG sites were identified from Direction1, particularly for IFN-γ and IL-1β, indicating the important role of long-term epigenetic modifications in regulating changes in both monocytes and T-cell-derived cytokine production. For instance, cg25926804, located near the *TMEM75*

and *GSDMC* gene, mediated 28.6% of the effect of rs6991078 on TI (IFN-γ) (Fig. 4D). *GSDMC* is known to be involved in the antimicrobial defense response [36, 37]. Similarly, cg21375332 (near *SLC12 A3* and *NUP93*) mediated 32.7% of the effect of rs604639 on IFN-γ changes (Fig. 4E). *SLC12 A3* is a receptor for the pro-inflammatory cytokine IL-18 and was reported to contribute to IL-18-induced cytokine production, including IFN-γ,,IL-6, IL-18, and CCL2 [38]. Interestingly, we also found cg19226099, located in *MC3R,* to act as a mediator for IFN-γ (Fig. 4F). MC3R is the receptor for melanocyte-stimulating hormone (MSH) and adrenocorticotropic hormone (ACTH) was reported to be related to a delay in the age of puberty onset [39], suggesting the importance of DNA methylation changes in genes related to hormones that mediate the effect of cytokine production changes after BCG vaccination. In the results from Direction2, where the trained immunity was treated as a mediator, few results were identified compared to Direction1, including cg07420470, located near *RAB3GAP2*, which was mediated by TI (IFN-γ) (Additional file 1: Fig. S9). *RAB3GAP2* is involved in the regulated exocytosis of neurotransmitters and hormones [40].

### Sex-specificity of BCG effect on DNA methylation and the relationship between DNA methylation and immune responses

In a previous study [9], sex-specificity in BCG-induced inflammatory protein changes was observed, indicating that the BCG vaccination might act differently in men and women. To further investigate this phenomenon, we stratified our samples by sex and assessed the short- and long-term changes separately in males and females, utilizing the same model as in the previous analysis. Our analysis revealed that following BCG vaccination, there were suggestive significant short-term changes in 18 CpG sites among males and 31 CpG sites among females ($P < 1 \times 10^{-5}$, Additional file 2: Table S21) without any overlap, and most of the identified CpG sites show different directions of changes indicating a strong sex-specificity (Fig. 5A–B). Specifically, at T14 compared to T0, we observed 13 demethylated sites (72.2%) in males and 10

(See figure on next page.)

**Fig. 5** Sex-specific effect on BCG vaccination and the association with ex vivo cytokine production changes. **A** Miami plot showing the short-term DNA methylation changes upon BCG vaccination (T14–T0) in females (upper) and males (lower). CpG sites with $P < 1 \times 10^{-5}$ were highlighted by red (female) and blue (male). **B** Scatter plot showing the consistency of results from males and females regarding the short-term effect. The *x*-axis represents the $-\log10$ *P* value with the sign of change direction in females, and the *y*-axis showing the same value in males. The dots are significant CpG sites identified from female (red dots) and male (blue dots) respectively with $P < 1 \times 10.^{-5}$. (C) Miami plot showing the long-term DNA methylation changes upon BCG vaccination (T90–T0) in females (upper) and males (lower); the only CpG site that passed FDR significance was highlighted by green. **D** Scatter plot showing the consistency of results from males and females regarding the long-term effect. **E**–**F** Dot plot showing the pathway enrichment of genes annotated to the identified CpG sites in short-term (**E**) and long-term (**F**), with two panels for females (left) and males (left) respectively. **G** Miami plot showing the association between baseline DNA methylation and ex vivo production changes of IFN-γ in females (upper) and males (lower) with FDR < 0.05. **H** Scatter plot showing the consistency of the results from G in males and females. The *x*-axis represents the $-\log10$ *P* value with the sign of the correlation coefficient in females, and the *y*-axis showing the same value in males. **I** Heatmap showing the Spearman correlation between TI (IFN-γ) associated CpG sites and baseline hormone levels of androstenedione (ADNC), cortisol (CORC), and 11-deoxycortisol (DESC). Only significant correlations (FDR < 0.05) are shown in this heatmap. Cell colors indicate the $-\log10(P)$ with the sign of the correlation coefficient of Spearman correlation (rho), and the asterisks indicate the significance of the correlation (* FDR < 0.05, ** FDR < 0.01)

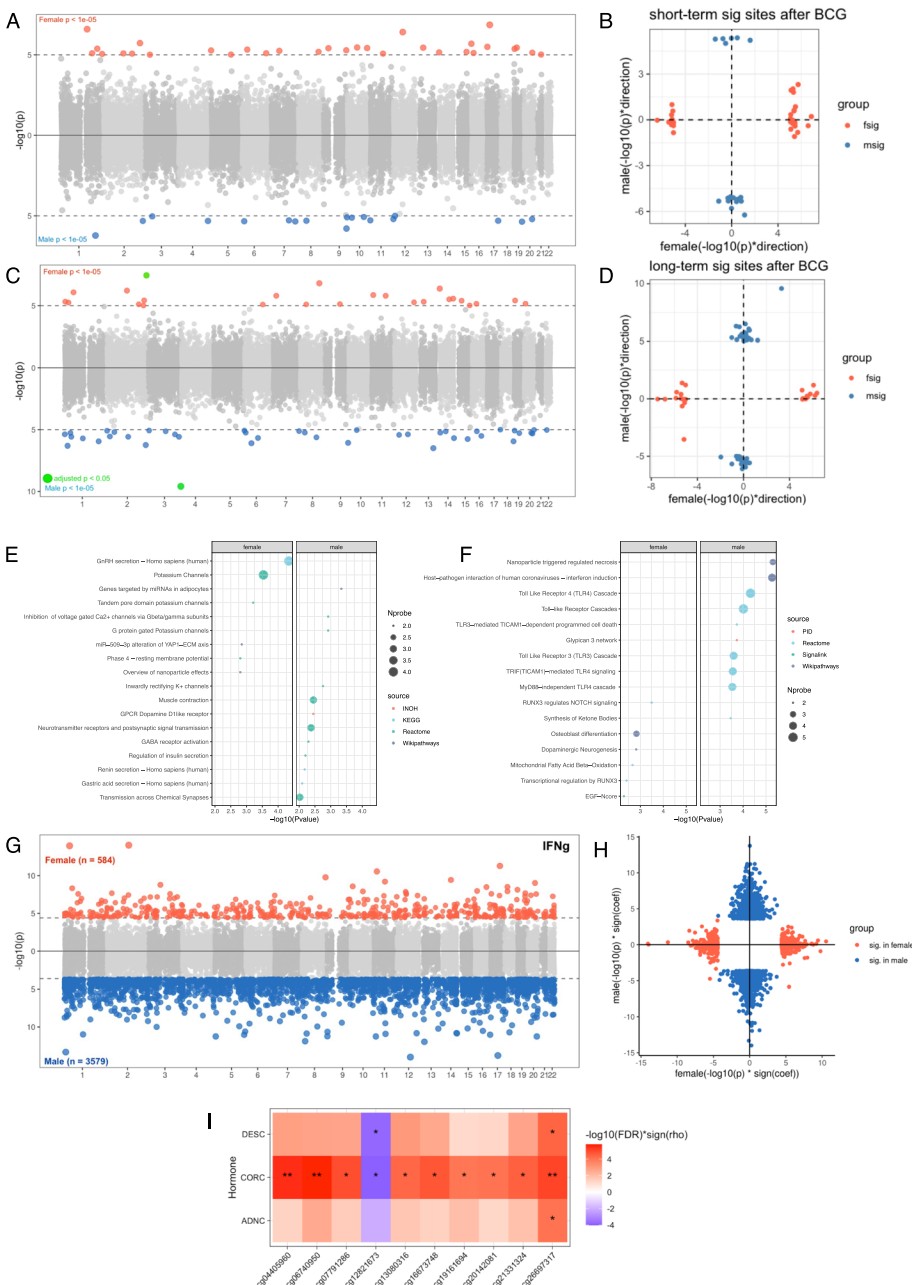

**Fig. 5** (See legend on previous page.)

demethylated sites (32.3%) in females. Through enrichment analysis, we identified sex-specific pathways associated with these methylation changes. Among females, the sex-hormone related pathway GnRH secretion (also modulated by kisspeptins, see above) displayed significant involvement, while among males, functions related to the nervous system, including neurotransmitter receptors and postsynaptic signal transmission, GABA receptor activation, and Renin secretion, were found to be prominent. Additionally, we identified a few shared pathways between both sexes, such as those related to potassium channels (pathway potassium channels in females, and pathway

G protein gated potassium channels and Inwardly rectifying K+ channels in males) (Fig. 5E).

We also identified suggestive significant changes in 42 CpG sites in males, and 25 CpG sites in females as long-term effects after vaccination ($P < 1 \times 10^{-5}$, Fig. 5C–D, Additional file 2: Table S22). Unlike the short-term BCG effect sites, the long-term changes in CpG sites exhibited the most consistent increases or decreases in males, whereas no consistent patterns were identified in females (Additional file 1: Fig. S10). Pathway analysis revealed that the CpG sites showing long-term effects in males were enriched in functions related to infection and immune responses (host − pathogen interaction of human coronaviruses and Toll − like Receptor pathways) and cell death (nanoparticle triggered regulated necrosis). In contrast, the long-term sites in females were enriched in pathways including RUNX3 regulates NOTCH signaling, dopaminergic neurogenesis, and osteoblast differentiation (Fig. 5F). These findings indicate that after BCG vaccination, the DNA methylation profiles undergo different trajectories in males and females, and these BCG-induced methylation changes are associated with distinct functional pathways. In summary, the short-term changes in methylation following BCG vaccination were enriched to functions related to sex hormones in females and the nervous system in males, while the long-term changes were associated with pathways related to pathogenic immune responses in males and the nervous systems in females.

The ex vivo cytokine production changes were reported to be associated with inflammatory proteins in a sex-dependent manner [9]. We also observed the sex-specific association between trained immunity of four cytokines and baseline epigenetic profiles by sex-stratified analysis, with a stronger association identified in males than females for all four cytokines tested in this study. In males, we identified 3579 CpG sites at baseline that were significantly associated with TI (IFN-γ) males, whereas only 584 sites showed significant association in females, without any overlap (FDR < 0.05, Fig. 5G, Additional file 2: Table S23). Furthermore, the direction of 1883 out of the total 4161 (45.3%) associations was inconsistent between males and females (Fig. 5H). Similar trends were observed for other cytokine markers, such as IL-1β (1107 in males and 62 in females), IL-6 (853 in males and 67 in females), and TNF-α (513 in males and 9 in females) (Additional file 2: Table S24). Importantly, these associated sites were not enriched to the CpG sites that were significantly associated with sex at baseline (Fisher's exact test $P > 0.05$, Additional file 2: Table S25). Instead, pathway analysis revealed that CpG sites associated with TI (IFN-γ) in both males and females were enriched in signal transduction pathways. However, other enriched pathways were specific to males or females. For instance, the top pathways in males included signaling by receptor tyrosine kinases, the Rap1 signaling pathway, and pathways in cancer, while in females, the top pathways were the Hippo signaling pathway, dopaminergic neurogenesis, and NrCAM interactions (Additional file 1: Fig. S11). To gain insight into the sex specificity of these associations, we also associated the identified CpG sites with five types of sex hormones including androstenedione, cortisol, 11-deoxycortisol, 17-hydroxyprogesterone, and testosterone. Interestingly, 10 CpG sites associated with TI (IFN-γ) in females were strongly associated with cortisol, and two CpG sites were also associated with 11-deoxycortisol (Spearman correlation FDR

< 0.05, Fig. 5I), while for the sites identified in males, we did not identify any significant links with sex hormones.

Due to the limited number of male samples collected available ($N = 4$) in the replication cohort, we were only able to validate the female-specific results. We first tested the long-term female-specific CpG sites ($N = 25$) comparing the difference of DNA methylation levels between T90 and T0, and 12 out of 25 CpG sites showed consistent directions of changes with one CpG site reaching nominal significance ($P < 0.05$, Additional file 2: Table S26). Additionally, for the female-specific associations between baseline DNA methylation and TI (IFN-γ), we found that 34 out of 582 CpG sites can be replicated with nominal significance ($P < 0.05$) and consistent directions of effect. Among these, 10 CpG sites reached the FDR significance (FDR < 0.05, Additional file 2: Table S27).

## Discussion

Epigenetic reprogramming has been proposed as a mechanism underlying BCG-induced innate immune memory. In this study, we provide evidence for the dynamic landscape of DNA methylation changes following BCG vaccination and elucidate the role of DNA methylation in immunological responses induced by BCG vaccination. Our findings demonstrate that BCG induces long-term immune-related DNA methylation changes for up to at least 3 months, likely reflecting epigenetic memory at the DNA methylation level (Fig. 6). Our results reveal sex-specific effects of BCG-induced DNA methylation changes and cytokine production, highlighting the importance of considering sex differences in immune response studies.

While long-term epigenetic processes have been proposed earlier to mediate trained immunity, most studies to date have investigated histone modifications [41, 42]. A recent study investigated changes in DNA methylation in monocytes of young children [20], while we suggested that DNA methylation changes after BCG vaccination in adults as well [22]. A comprehensive analysis of the impact of DNA methylation on innate immune memory in adults was missing. In this study, we observed distinct patterns of

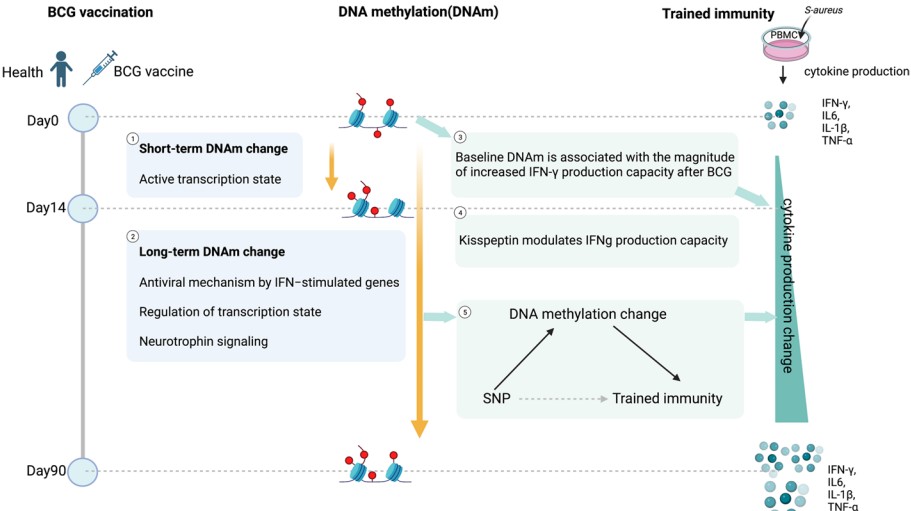

**Fig. 6** Summary of findings of this study. Figure was created using BioRender.com

short- and long-term DNA methylation changes following BCG vaccination. Previous research has outlined distinct stages of epigenetic reprogramming in the induction of trained immunity, involving acute stimulation of innate immune cells leading to active transcription of proinflammatory factors, followed by a resting stage, where epigenomic changes are partially reversed after stimulus cessation [43]. In our study, we observed similar acute effects on DNA methylation at day 14 that returned back to baseline by day 90 following BCG vaccination. Notably, the demethylated CpG sites associated with this acute effect were enriched in an active transcription state, consistent with the previous finding that BCG vaccination induced acute and resting stages. Besides, the long-term epigenetic changes, potentially contributing to epigenetic memory, were enriched in infection-related pathways and associated with ex vivo cytokine changes that represented innate immune memory. These findings suggest that epigenetic memory partly underlies the induction of immune memory. Furthermore, we identified a "late" epigenetic reprogramming that includes decreased hypermethylation and increased hypomethylation. DNA methylation is a known stable biomarker of aging, with a tendency towards increased hypermethylation in aging processes [44]. Interestingly, our results suggest that the "late" effect of BCG vaccination opposes the DNA methylation changes associated with natural aging, indicating a distinct epigenetic influence of BCG vaccination.

BCG vaccination has been shown to induce epigenetic reprogramming, such as DNA methylation [45] and histone modifications [2], which play a critical role in trained immunity or immunological memory. In our study, we investigated the association between DNA methylation levels and immunological memory following BCG vaccination, specifically focusing on ex vivo cytokine production changes of four cytokines upon *S. aureus* stimulation. We found that baseline DNA methylation explained a significant proportion of variance of T-cell heterologous IFN-γ responses, and this may also play a role in the process of immune memory [46]. These results indicated that individual differences in long-term immune effects induced by vaccines depend on intrinsic DNA methylation levels prior to vaccination, and DNA methylation status at certain CpG sites may facilitate the immune responses after BCG vaccination. Among the pathways modulated by DNA methylation that were found to strongly modulate heterologous immune responses, we identified baseline CpG sites enriched in the kisspeptin pathway. Kisspeptins are peptides that induce the release of GnRH, but with an increasing number of studies showing immunomodulatory effects [47, 48]. Indeed, functional validation experiments of kisspeptin demonstrate their modulatory effects on IFN-γ production, strengthening the hypothesis of their impact on long-term immune responses.

Moreover, we identified that BCG-induced DNA methylation changes at day 90 compared to baseline, and these were associated with ex vivo cytokine changes including both IFN-γ and IL-1β, representing responses from both monocytes and T-cell-derived cytokine responses. Previous studies have shown that the stimulation of innate immune cells can leave an "epigenetic scar" making the cells more responsive to different stimulation [43]. Our results suggest that epigenetic memory is represented by a combination of innate immune training and heterologous adaptive immune memory [13, 49]. Previous research has indicated that IFN-γ plays a critical role in amplifying the trained immunity response at the transcriptomic level (13). Our findings may suggest that crosstalk

between innate immune cells and adaptive immune cells may be crucial for the development of the trained immunity at the epigenetic level. Furthermore, through mediation analysis, we showed the potential in silico causal relationship between BCG-induced long-term methylation changes and trained immunity. We identified more CpG sites that may mediate the regulation of genetic variants on IFN-γ and IL-1β changes after BCG vaccination, suggesting the DNA methylation acts as a modulator of immune response. The identified CpG sites were located in genes related to pathogenic responses and cytokine productions, which stressed the importance of epigenetic reprogramming at CpG sites on the regulation of trained immunity. For example, the DNA methylation changes of cg21375332 near *SLC12 A3* gene mediated the regulation of genetic variants on IFN-γ changes, and this was consistent with a previous study that *SLC12 A3* was involved in IL-18-induced IFN-γ production [38]. We also revealed that the CpG site near *GSDMC*, a gene involved in antimicrobial defense response, mediated the changes of IFN-γ after BCG vaccination.

Several studies have demonstrated that the BCG vaccination can exhibit sex-specific effects [50, 51]. In a study investigating the protective effects of neonatal BCG vaccination, it was observed that boys exhibited strong protective effects within the first week after vaccination, which diminished thereafter. Conversely, girls showed weaker protection initially but demonstrated stronger effects later on [51]. In our study involving adult participants, we also identified sex-specific DNA methylation changes induced by BCG vaccination, both in the short and long term. Short-term DNA methylation changes in females were enriched to the GnRH pathway, which influences hormone secretion and immune system development [52]. Long-term effects in males involved DNA methylation changes related to Toll-like receptors (TLRs), particularly TLR4, which recognizes bacteria lipopolysaccharide (LPS) and activates innate immunity. Conversely, long-term effects in females were observed in CpG sites associated with neuronal functions, including Notch signaling and dopaminergic neurogenesis. It is worth noting that the neonatal BCG vaccination was reported to improve neurogenesis in mice [53].

Moreover, baseline DNA methylation levels impact the cytokine changes following BCG vaccination in a sex-specific manner. We identified the dopaminergic neurogenesis pathway in females to influence the IFN-γ associated CpG sites, which confirmed the importance of epigenetic modifications in the neuronal crosstalk with the immune system after BCG vaccination in females. We also revealed that the IFN-γ associated CpG sites in males were enriched in the receptor tyrosine kinases pathway, which was known as key regulators of an uncontrolled immune response and played a critical role in the control of autoimmune disorders[54]. Interestingly, we observed an association between IFN-γ associated CpG sites and cortisol, specifically in females. Cortisol, a mediator of stress-induced immune-suppression, may affect both innate and adaptive immunity. A previous study revealed that infants with higher cortisol response to pain and lower delayed-type hypersensitivity response to BCG vaccination [55].

This study has some limitations that should be acknowledged. First, DNA methylation profiles were measured from whole blood samples. Since DNA methylation measurements are cell-type specific, the use of whole blood samples may mask the cell-type specific methylation changes. Second, the DNA methylation analysis conducted using the EPIC array only covers approximately 3% of the CpG sites in the genome. Third, while

we were able to partially replicate our results in an independent cohort, the validation study had limited statistical power due to the limited sample size. Further validation of the association between BCG vaccination-induced methylation change and trained immunity is still needed. Fourth, we did not identify genome-wide significant CpG sites associated with BCG effect in male and female subgroups (FDR < 0.05), and thus the sex-specific analyses are restricted by the relatively small sample size and the male-specific results lack external validation. Fifth, we were not able to assess the transcription of the genes influenced by the CpG sites at the expression level, which limits the interpretation of our results. Sixth, this study only assessed the role of BCG-induced DNA methylation changes in ex vivo cytokine production through association and mediation analyses; further functional studies are still needed to explore the molecular mechanism. Seventh, the study population consisted primarily of young adults (< 30), and the generalizability of our findings to older people and the age-different methylation patterns induced by BCG vaccination still deserves further investigation. Finally, while our study provides evidence for the role of BCG-induced DNA methylation changes in cytokine production, alternative explanations should be considered. The lack of a non-vaccinated control group may limit attributing the observed changes to BCG-induced trained immunity.

## Conclusions

In conclusion, our study provides valuable insights into the dynamic changes of DNA methylation following BCG vaccination and highlights the induction of immune-related epigenetic memory through these dynamic DNA methylation changes. The observed DNA methylation changes play a regulatory role in immunological responses triggered by BCG vaccination, underscoring the presence of immune-related epigenetic memory.

## Methods

### Study design and cohort description

In the 300BCG study, 303 healthy Dutch individuals were included from April 2017 until June 2018. After obtaining the written informed consent, blood was collected, followed by the administration of a standard dose of 0.1 mL BCG (BCG-Bulgaria, InterVax) intradermally in the left upper arm by a medical doctor. Two weeks and 3 months after BCG vaccination, additional blood samples were collected and a questionnaire was completed. Exclusion criteria were the use of systemic medication other than oral contraceptives or acetaminophen, use of antibiotics 3 months before inclusion, previous BCG vaccination, history of tuberculosis, any febrile illness 4 weeks before participation, any vaccination 3 months before participation, or a medical history of immunodeficiency. This study was approved by the Arnhem-Nijmegen Medical Ethical Committee (NL58553.091.16).

### DNA methylation measurement and quality control

DNA was isolated from frozen whole blood using a QIAamp blood kit (Qiagen Benelux BV, Venlo, the Netherlands) kit, and the concentration was determined using a NanoDrop spectrophotometer at 260 nm. Genome-wide DNA methylation profiles were measured by Infinium© MethylationEPIC array (~ 850,000 CpG sites). DNA methylation data were pre-processed in R (version 4.0) with the Bioconductor package Minfi [56], using the original IDAT files extracted from the HiScanSQ scanner. We removed

the samples which had bad call rate ($n = 4$), sex-mismatched ($n = 1$), and we checked whether we had mixed-up samples by inspecting the correlation of the beta value for SNPs. Mismatched samples were replaced by a new label with a correlation of larger than 0.7. Quality control was performed to filter bad quality probes with a detection *P*-value > 0.01, cross-reactive probes, polymorphic probes [57, 58], and probes in the sex chromosome. We subsequently implemented stratified quantile normalization [59]. Based on methylation value, cell proportion was estimated using Housman's method [24]. After quality control and matching with phenotype data, 284 samples and 751,564 probes remained for further analyses. Methylation levels (beta values, $\beta$) were converted into $M$ values ($\log2(\beta/1 - \beta)$), which were used in downstream differential methylation analyses. Extreme outliers in the methylation data were identified using the Tukey method (< 1st quartile $- 3 \times$ IQR; > 3rd quartile $+ 3 \times$ IQR) and set as missing.

### Assessment of ex vivocytokine responses

Peripheral blood mononuclear cells (PBMCs) were isolated from EDTA whole blood with Ficoll-Paque (GE Healthcare) density gradient separation, immediately after collection. Cells were washed twice in PBS and suspended in RPMI culture medium (Roswell Park Memorial Institute medium, Invitrogen, CA, USA) supplemented with 50 mg/mL gentamicin (Centrafarm), 2 mM glutamax (GIBCO), and 1 mM pyruvate (GIBCO). Subsequently, the PBMCs were stimulated ex vivo with $10^6$ CFU/mL heat-killed *Staphylococcus aureus* or left unstimulated. Cytokine production of TNF-α, IL-6, and IL-1β was measured in 24-h supernatants, and IFN-γ was measured in supernatants 7 days after stimulation using ELISA. Then, we calculated the fold change in cytokine production (3 months after vaccination compared to baseline). After log10 transformation and correcting for batch effect, the value was used as a measurement of the magnitude of the trained immunity response. For the in vitro validation of the immunomodulatory role of kisspeptin, a similar 7-day stimulation assay for the production of IFN-γ was employed in human PBMCs.

### Statistics

Statistical tests were conducted using R (version 4.0.4 (www.r-project.org). A *p*-value < 0.05 after multiple testing with FDR was considered statistically significant. A suggestive significance threshold of $p < 1 \times 10^{-5}$ was applied for the exploratory analysis to identify overall consistent patterns. The statistical inflation of all the EWAS results is provided in Additional file 2: Table S28.

### Association of DNA methylation with BCG effect and immune responses

Principle component analysis (PCA) was performed using DNA methylation data from each time point respectively. The top 30 PCs were associated with variables including age, sex, batch (sample plate), BMI, smoking, and estimated cell counts by linear regression model.

To assess the DNA methylation changes induced by BCG vaccination, a linear mix effect model was performed for each CpG site, with time point as the independent variable, subject ID as the random effect, and age, sex, batch (sample plate), and estimated cell counts as the covariates. Besides the overall changes, we also assessed the changes

from T0 to T14, T0 to T90, and T14 to T90, using the same model which only included two time points. EWAS of trained ex vivo cytokine response at each time point was performed by fitting the robust linear regression model adjusting for age, sex, batch (sample plate), and estimated cell counts. The ex vivo cytokine response was normalized by the inverse rank method. Besides, we also assessed the association between DNA methylation changes and ex vivo cytokine production changes using the robust linear regression model. Considering the different estimated cell counts at different time points, we first regress out the cell counts and batch (sample plate) at each time point from DNA methylation M values, and the residuals were obtained. Then we used a robust linear regression model to correlate the DNA methylation changes (differences in the residuals at any two time points T90–T0, T14–T0, and T90–T14), to the trained immunity response with age and gender as covariables. CpG sites passed the threshold of FDR < 0.05 were considered significant findings. The sex-stratified analysis was performed in male and female respectively using the same models mentioned above.

The identified CpG sites were annotated to nearby genes by the GREAT annotation tool [60]. A tool called experimentally derived Functional element Overlap analysis of ReGions from EWAS (eFORGE) 2.0 [27] was used to annotate the CpG sites associated with the BCG effect to further understand the function of these CpG sites in transcription activity. We applied this tool to methylated and demethylated CpG sites with $P$ value $< 10^{-5}$, respectively, of the overall BCG model. Pathway analysis of genes mapped to identified CpG lists was performed with the online tool CPDB http://cpdb.molgen. mpg.de/. DNA methylation levels and estimated cell proportions at different time points (for 11 CpG sites associated with the overall BCG effect) were performed by Spearman correlation. Neutrophil cell counts were also measured by flow cytometry, where the sub-cell types were also measured. And then the cell counts were correlated to 11 CpG sites by Spearman correlation.

The assessment and quality control steps for inflammatory protein data were published previously [9]. Proteins that were significantly decreased at T14 compared to T0 ($N = 24$), and T90 compared to T0 ($N = 10$) were selected based on the previous study [9]. CpG sites that were significantly changed at T14 compared to T0 ($N = 39$), and T90 compared to T0 ($N = 36$) were selected by previous steps in this study. The associations between DNA methylation changes (T14–T0 and T90–T0) and inflammatory protein changes (fold change T14/T0 and T90/T0) were performed by Spearman correlation.

### Sensitivity analyses

Considering that cell type heterogeneity can be a significant confounder in the EWAS study, we also applied EpiDISH [28], which predicts the proportions of 12 cell types, to achieve a higher resolution of blood cell composition. Sensitivity analyses were performed on the top CpG sites associated with BCG effect (changes in DNA methylation over time after BCG vaccination, $P < 1 \times 10^{-5}$) and TI (baseline DNA methylation and cytokine productions, including IL6 and IFN-γ, $P < 1 \times 10^{-5}$).

To evaluate whether the significant CpG sites identified in our study are influenced by the seasonality, we performed two additional analyses: 1) For the BCG-associated CpG sites, we included seasonality as an additional covariate in the model; 2) We assessed whether top CpG sites (BCG-associated and TI-associated) were correlated with

seasonality at baseline. The seasonality was calculated based on the method described in a previous study [61].

Considering the study population is mostly young adults under 30 years old, we further performed sensitivity analyses in the subgroup of participants older than 30 years old for the BCG-associated CpG sites, using the same model as the original analyses.

### Variance explanation

We estimated the variance explained by age, sex, PRS, as well as DNA methylation, inflammatory proteins, and circulating metabolites at baseline, on ex vivo IFN-γ response. The PRS was calculated from SNPs associated with the trained immune response. Inflammatory proteins at baseline were assessed by Olink inflammation panel [9]. Baseline metabolites were measured and annotated by the General Metabolics (Zurich, Switzerland) using flow injection time-of-flight mass (flow-injection TOF-M) spectrometry [10].

We first select the features by associating the features in each data level to TI (IFN-γ). If a feature showed a significant association (Spearman correlation, *P* value < 0.05), the feature was included as the candidate predictor. For DNA methylation, we included the 41 CpG sites identified by EWAS analysis. For metabolites data, we included 34 metabolites that were significantly associated with TI (IFN-γ, $P < 0.05$). And for protein data, we do not identify any significant proteins that were associated with TI (IFN-γ). Each candidate predictor from each data layer was correlated to other predictors within the data layer to identify collinearity among these predictors. If the features within this layer showed an association (Spearman correlation > 0.4), the feature which showed the least association (based on the *p*-value) to TI (IFN-γ) was further removed from the candidate predictors. This yielded a unique set of predictors from each layer, which was then used to fit a multivariate linear model to estimate the variance explained by these features for TI (IFN-γ). Finally, 31 CpG sites and 23 metabolites were selected by these approaches and were used in the variance calculation. To account for the inflation that adding more predictors has on the explained variation, the adjusted $R^2$ was used as the measure of explained variance. Additionally, we also randomly selected the same number of features ($n = 73$, the number of proteins available in this study) 1000 times and repeated the approaches mentioned above to get the variance explained by randomly selected features.

### Cytokine-QTL and enrichment analysis

The association between SNP dosages and IFN-γ responses (cytokine QTL) was done using the Matrix eQTL R package [62] with the inversed rank normalized IFN-γ responses value as the dependent factor, adjusting for age and sex. Using the identified IFN-γ associated CpG sites list ($N = 41$, Table S13), cytokine QTL *p*-values of those SNPs located 250 kb upstream or downstream of those CpG sites were extracted and plotted against random cytokine QTL *p*-values. By this, the *cis* effect of CpG sites on cytokine QTL was tested.

### Bi-directional mediation analysis

Matrix eQTL [62] was used to identify the significant pairs of TI-SNPs and DNAm-C SNPs after regressing out age and gender. For SNPs significantly associated with both TI and DNAm-C ($P < 0.05$), we carried out bi-directional mediation analysis ($y = x + m + \varepsilon$, where $y$ is the outcome, $x$ is the SNP dosage and $m$ represents the mediator) using the mediation R package (ref mediation: R Package for Causal Mediation Analysis) to infer the mediation effect of TI or DNAm-C for genetic impacts.

### Replication cohort

We replicated our results in a cohort consisting of 17 individuals from the BCG booster cohort [23]. This randomized placebo-controlled trial originally compares different BCG vaccination regimens for identifying their efficacy to establish trained immunity. The 17 volunteers that received one dose of BCG as positive control were used as validation cohorts in the current study. Vaccination was performed intradermally in the left upper arm in the same manner as with the discovery cohort. Blood was drawn at baseline and 3 months after the vaccination. The trial protocol was registered under NL58219.091.16 in the Dutch trial registry and was approved in 2019 by the Arnhem-Nijmegen Ethics Committee. All experiments were conducted in accordance with the Declaration of Helsinki, and no adverse events were recorded.

### Supplementary Information

Additional file 1: Supplementary figures Fig.S1-11.

Additional file 2: Supplementary tables Table S1-28.

Additional file 3: Review history.

**Acknowledgements**
We would like to thank all volunteers from the 300BCG cohort for participation in the study. We thank BioRender.com for providing the icons used in the Figure 1.

**Review history**
The review history is available as Additional file 3.

**Peer review information**

**Author contributions**
MGN, CX, and YL conceptualized and designed the study. CQ and ZL performed the data analysis supervised by YL, CX, and MGN. SJCFMM and VACMK recruited the participants and collected the biological material. GK, ASS, and PAD performed DNA isolation and supported the functional experiments; helped with participant recruitment and interpretation of the data, with support from AP, LCJdeB, VPM, and LABJ. YAM, WL, MG, and AA helped with part of the data analysis. CQ, ZL, CX, and MGN wrote the manuscript with input from all the authors. All authors reviewed and approved the manuscript.

**Funding**
 YL was supported by an ERC starting Grant (948207) and a Radboud University Medical Centre Hypatia Grant (2018). CJX was supported by Helmholtz Initiative and Networking Fund (1800167) and Deutsche Forschungsgemeinschaft (DFG) Fund (497673685). MGN was supported by an ERC Advanced Grant (833247) and a Spinoza Grant of the Netherlands Organization for Scientific Research. CQ was supported by the National Natural Science Foundation of China (82302610).

**Data availability**
DNA methylation data have been deposited at the European Genome-phenome Archive (EGA), which is hosted by the EBI and the CRG, under accession number EGAS00001007498 [63].
Code generated to process the data are freely available on Github (https://github.com/CiiM-Bioinformatics-group/BCG_methylation_project) [64], and on Zenodo (https://doi.org/10.5281/zenodo.14904253) [65] under an MIT license.

## Declarations

### Ethics approval and consent to participate
The 300BCG study was approved by the Arnhem-Nijmegen Medical Ethical Committee (NL58553.091.16). All participants have given written informed consent.

### Competing interests
MGN is the scientific founder of TTxD, Lemba, Salvina, and Biotrip. LABJ is the scientific founder of TTxD, Lemba, and Salvina. None of these start-ups have objectives that intersect with this study. The other authors declare that they have no competing interests.

### Author details
[1]Centre for Individualised Infection Medicine (CiiM), a joint venture between the, Helmholtz Centre for Infection Research (HZI) and Hannover Medical School (MHH), Hannover, Germany. [2]TWINCORE, a joint venture between theHelmholtz-Centre for Infection Research (HZI)Hannover Medical School (MHH), Hannover, Germany. [3]Microbiome Medicine Center, Division of Laboratory Medicine, Zhujiang Hospital, Southern Medical University, Guangzhou, Guangdong, China. [4]Department of Internal Medicine and Radboud Center for Infectious Diseases, Radboud University Medical Center, Nijmegen, The Netherlands. [5]Institute for Bioinformatics, University Medicine Greifswald, Greifswald, Germany. [6]Research Centre Innovations in Care, Rotterdam University of Applied Sciences, Rotterdam, The Netherlands. [7]Department of Medical Genetics, Iuliu Haţieganu University of Medicine and Pharmacy, Cluj-Napoca-Napoca, Romania. [8]Department of Immunology and Metabolism, Life and Medical Sciences Institute (LIMES), University of Bonn, Bonn, Germany.

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

## 