## [Additional file 3: Review history. · Genome Biology]

Review history

First round of review

Reviewer 1

Qi and colleagues performed an analysis of DNA methylation in whole blood of 300 subjects vaccinated with BCG and follow DNA methylation changes and ex vivo cytokine production after 14 and 90 days. The study design of substantial interest and the dynamics of DNA methylation responses over time is of interest to the field. Instead of correcting for multiple testing, the authors use a p-value threshold for suggestive significance. While uncommon, it can be justified in this particular case where longitudinal data is present and a clear case is made for consistency of results. After the main analysis the presentation of results, a wide range of follow-up analyses is presented among which analysis that feel ad hoc with speculative interpretation. Given that BCG vaccination responses is framed as a model for trained immunity, which the authors have reported to be mediated by the innate immune system, the analysis of whole blood samples is a limitation. A simplified and more focused presentation may be able to highlight the value of the findings.

- The title contains claims that are too strong and this returns throughout the manuscript. The authors write that the novelty of their study is that DNA methylation is assessed and no other levels of epigenetic information are interrogated. There is no evidence presented for reprogramming since changes were observed at 1 time-point only. Also the functional impact is inferred based on the association of the nearest genes with a diverse array of processes. There is much less indication for a role of monocyte responses than T cell responses since associations with INF-gamma stand out. Also the word mediation (=causation) is too strong a claim since it is based on statistical significance for a subset of mediation analyses and no evidence for a causal, mechanistic role for DNA methylation is presented. A more accurate phrasing of the title and throughout the manuscript is for example 'DNA methylation changes are associated with heterologous T cell-derived responses after BCG vaccination'.
- The main analysis is corrected for cell counts, but only the older prediction of 6 cell types. The authors should include the current 12-cell count prediction as for example implemented in EpiDISH preferably as main analysis or alternatively as sensitivity analysis to filter out affected CpGs. Clearly, a limitation of the study is the analysis of whole blood while the responses of specific immune cells have been reported for BCG vaccination, in particular of the innate immune system. The analysis of 12 cell types may lead to more insight.
- A surprising aspect of the manuscript is that replication is sought only after all analyses were done at the very end of the paper. The order should be reversed and replication should be tested immediately after the discovery phase. Given the relatively small replication set, this analysis can focus on consistency rather than formal statistical significance (e.g. direction of effects). The change in order is particularly important because the authors rely on a lenient threshold for suggestive statistical significance.
- The authors now only report scaled DNA methylation differences (e.g. figure 2). While this looks visually convincing, it removes the opportunity to gauge the actual effect sizes and their interpretation. It is important to (also) show the differences as methylation fractions (beta-value scale; which may for example be set relative changes to those at t=0). This is relevant to know to what extent the DNA methylome is changed and could be of functional relevance.
- The authors should report and, if necessary, address statistical inflation of test statistics.
- The discovery analysis includes batch as covariate. Please, specify which batch (sample

plate?).

- A very intriguing finding is that the study highlights the role of T cell responses in BCG vaccination and downplays the responses of monocytes. This would have been missed if monocytes were studied instead of whole blood. Did the authors consider analysis like interaction tests to gain some indication in which cell type the DNA methylation difference occurred? Are transcriptomic data available? This could very much strengthen the insight in and evidence for a functional effect of the DNA methylation differences observed and - e.g. using cell type specific expression patterns - could pinpoint the putative cell type involved. Irrespectively, the manuscript will be improved by a discussion of the various alternative explanations. More discussion highlighting the involvement of T cells in contrast to innate immune cells is justified. In contrast, the discussion mentions 'innate immune cells' and 'epigenetic scar'. Given that whole blood is studied and associations were found for T cell responses, this conclusion cannot be founded on the data.

- The link with aging put forward in the discussion is superficial.

- After the main analysis, a series of follow-up analysis are presented which are not all performed with similar rigor or clear research question.

> DNA methylation changes are associated with protein levels. Some associations are found but the functional role and specific link with the outcomes of interest remain unclear.

Moreover, despite testing many proteins, associations with p-values that were merely <0.05 are pushed as of interest. Als the last sentence speculates on a causal role which is not supported by the type of analyses done nor supported by the outcome.

> Baseline DNA methylation is correlated to ex vivo cytokine production. This analysis does not link to the research question whether vaccination does induces a DNA methylation response. The number of CpGs associated with outcomes is mentioned, which is not informative. Some analyses are put forward as 'interestingly (line 286) but it remains unclear to the reader why this is interesting because not link with the research question is made. All in all, this analysis seems out of place.

> The kisspeptin result is potentially interesting but it is not clear where it comes from. Is it a candidate gene analysis? Why this one and not any of the many other potentially interesting ones? Where 'in this study' was an association with methylation observed?

> BCG-induced DNA methylation changes and their association with ex vivo cytokine production seem very much on topic. More context on the interpretation of associations and consistency of findings is warranted. Now mainly the number of associations per cytokine are mentioned. This is not particularly informative. Finally, earlier, the authors seem to find it interesting that most differentially methylated CpGs were observed for the T14-T19 comparison, but none of them was associated with ex vivo cytokine production. So, this is worrying?

> The mediation analysis is of potential interest, but due to inherent limitations cannot be simply interpreted as causal effects (and also not inform on the persistence of methylation differences).

> The sex-specificity analysis is not robust. Most main effects are not formally significant after correcting for multiple testing, this sub-group analysis is not supported by sample/effect sizes and results are at best speculative.

- The manuscripts now at times mentions that 'functional validation experiments' were done. Perhaps the ex-vivo cytokine production assays are meant. However, all analyses in the manuscript rely on association. This is not a bad thing, but should be phrased as such and described accordingly in the discussion.

- No non-vaccinated controls (placebo) were included in the study. Could seasonal or other

effects have contributed to findings?

- The authors study the BCG response. It is not clear why the results can be generalized to the concept of trained immunity. This requires more support or toning down. The apparent limited involvement of innate immune cells should be considered. Of note, the second paragraph of the introduction only refers to studies of BCG vaccination and no other exposures.
- The limitations paragraph in the discussion is overly optimistic and should provide a complete overview of limitations and alternative explanations for results should be considered.
- The authors use the term persistence. This term should be removed. The authors show that the changes at day 14 are not persistent and for the changes at day 90 we simply do not know what happens later.
- Page 6, line 198: 'the methylated CpG sites' not immediately clear this refers to an increase in methylation in contrast to demethylation.
- The existence of a short-term effect on DNA methylation that seems to be reversed from day 14 to 19 and replaced by a distinct longer-term DNA methylation signature is quite interesting. And also seems new. The only previous example I have seen in the literature from a different context is from Tilburg et al (Epigenomics 2020). Are there other examples?

Reviewer 2

The study investigates the role of DNA methylation in BCG-induced trained immunity using a substantial cohort of 303 healthy volunteers, providing good statistical power. Data is collected at multiple time points (T0, T14, T90), allowing for the assessment of both short-term and long-term effects. DNA methylation data is integrated with cytokine production and plasma inflammatory protein concentrations. The integration of genetic data and the mediation analysis between genetic variants, DNA methylation, and cytokine responses is a strength. Some of the weaknesses of the study are:

The use of whole blood DNA may mask cell-type-specific methylation changes. While the authors estimate cell proportions, direct measurement of cell populations or single-cell analysis would have provided more accurate and insightful data.

The study doesn't provide a clear mechanistic explanation for how the observed methylation changes contribute to trained immunity.

The authors mention 'monocyte and heterologous T cells' in the paper's title, which is not supported by the data. Although the correlation of methylation changes with neutrophil subtypes has been shown, no specific analysis of monocytes or T cells is presented.

The authors should focus the title on the broader findings related to DNA methylation changes in whole blood following BCG vaccination unless they have additional data specifically on monocytes and T cells.

The sex-specific effects observed in methylation changes and cytokine responses are interesting findings that could be highlighted more.

Some suggestions for change are:

* Only 11 CpG sites were identified as significantly changed (FDR < 0.05) following BCG vaccination, which is a relatively small number for a genome-wide study. The reason for this could be discussed.

* The authors could demonstrate how the observed methylation changes affect gene expression or cellular function, e.g., RT-qPCR for genes associated with the significantly changed CpG sites.

* The rationale for focusing on neutrophils could be explained.

* The manuscript reports a mean age of 25 years with a range of 18 to 71. The standard deviation for age should be mentioned. Age is a known confounder in DNA methylation studies.

Methylation patterns change with age, and these changes can be substantial. Without knowing the age distribution, it is difficult to determine if the observed methylation changes are truly due to BCG vaccination or if age differences influence them. If there's a wide age range, the effects of BCG vaccination on DNA methylation could differ between younger and older participants. Detailed age statistics, age-stratified analyses, and potentially exploring the age-vaccination interaction effects on DNA methylation changes could be explored.

* Sample handling details could be mentioned whether the blood samples were immediately processed to isolate DNA or was this done from frozen blood samples.

* A control group would be particularly useful for comparing baseline methylation values. This could help identify any pre-existing differences between the vaccinated and unvaccinated groups that might influence the results. A control group would further strengthen causal inferences about the effects of BCG vaccination. If the study period spans different seasons or other environmental changes, a control group could help account for these factors.

While the study has significant strengths and addresses an important topic, major revisions are needed to fully support the conclusions

Authors' response to reviewers

Reviewer #1: Qi and colleagues performed an analysis of DNA methylation in whole blood of 300 subjects vaccinated with BCG and follow DNA methylation changes and ex vivo cytokine production after 14 and 90 days. The study design of substantial interest and the dynamics of DNA methylation responses over time is of interest to the field. Instead of correcting for multiple testing, the authors use a p-value threshold for suggestive significance. While uncommon, it can be justified in this particular case where longitudinal data is present and a clear case is made for consistency of results. After the main analysis the presentation of results, a wide range of follow-up analyses is presented among which analysis that feel ad hoc with speculative interpretation. Given that BCG vaccination responses is framed as a model for trained immunity, which the authors have reported to be mediated by the innate immune system, the analysis of whole blood samples is a limitation. A simplified and more focused presentation may be able to highlight the value of the findings.

Authors' reply: We sincerely thank the reviewer for the thoughtful constructive feedback on our study. Regarding the use of the p-value threshold, we would like to clarify that we reported the FDR for each analysis. The p-value threshold was only applied in cases where the number of significant CpG site was small and when discussing overall patterns, to provide additional context for exploratory findings.

To address the reviewer's comments, we have added the following sentences to the manuscript:

“Statistics

Statistical tests were conducted using R (version 4.0.4 (www.r-project.org)). A p-value <0.05 after multiple testing with FDR was considered statistically significant. A suggestive significance threshold of $p < 1 \times 10^{-5}$ was applied for the exploratory analysis to identify overall consistent patterns.”

- 1. The title contains claims that are too strong and this returns throughout the manuscript. The authors write that the novelty of their study is that DNA methylation is assessed and no other levels of epigenetic information are interrogated. There is no evidence presented for reprogramming since changes were observed at 1 time-point only. Also the functional impact is inferred based on the association of the nearest genes with a diverse array of processes. There is much less indication for a role of monocyte responses than T cell responses since associations with INF-gamma stand out. Also the word mediation (=causation) is too strong a claim since it is based on statistical significance for a subset of mediation analyses and no evidence for a causal, mechanistic role for DNA methylation is presented. A more accurate phrasing of the title and throughout the manuscript is for example 'DNA methylation changes are associated with heterologous T cell-derived responses after BCG vaccination'.

Authors' reply: We thank the reviewer for this valuable suggestion. We agreed with the reviewer that only DNA methylation is assessed in the paper and no other levels of epigenetic information are interrogated. Therefore, we have changed the title to **“Long-term DNA methylation changes mediate monocyte and heterologous T cell-derived cytokine responses after BCG vaccination”**.

In this study, we applied mediation analysis, and the results indicate that DNA methylation changes mediate the IFN- γ response after BCG vaccination. However, we agree with the reviewer that here the mediation is not equivalent to causation. In this context, the term “mediation” refers to a statistical or in silico causality, as has been described in a previous study [PMID: 34847370]. To address this point, we have adjusted the phrasing throughout the manuscript to replace “causality” with “mediation effect”.

We also agree with the reviewer that associations with IFN- γ stand out, indicating a prominent role for T cell responses. However, we also observed associations between BCG-induced DNA methylation changes and increased production of IL-1 β upon *S. aureus* stimulation (Table S19 in the revised version). Since IL-1 β is predominantly produced by monocytes, this suggests that both monocyte and T cell responses may contribute to the observed outcomes.

To address the reviewer's comments, we have revised the manuscript as follows:

Page 13 Line 548-555

*“The majority of significant mediation results were identified for TI(IFN- γ). Specifically, seven CpG sites showed unidirectional **mediation effects** in Direction1 ($PD_{\text{Direction1}} < 0.05$ and $PD_{\text{Direction2}} > 0.05$), two CpG sites of unidirectional **mediation effects** in Direction2 ($PD_{\text{Direction2}} < 0.05$ and $PD_{\text{Direction1}} > 0.05$), and six CpG sites with bidirectional **mediation effects** ($P_{\text{mediation}} < 0.05$, Figure 4B, Table S20) ... along with four with bidirectional **mediation effects**, but none of the CpG sites were significant in Direction2 (Figure 4B, Table S20).”*

[PMID: 34847370]: Wang D, Doestzada M, Chen L, et al. Characterization of gut microbial structural variations as determinants of human bile acid metabolism. *Cell Host Microbe*. 2021;29(12):1802-1814.e5. doi:10.1016/j.chom.2021.11.003

- 2. The main analysis is corrected for cell counts, but only the older prediction of 6 cell types. The authors should include the current 12-cell count prediction as for example implemented in EpiDISH preferably as main analysis or alternatively as sensitivity analysis to filter out affected CpGs. Clearly, a limitation of the study is the analysis of whole blood while the responses of specific immune cells have been reported for BCG vaccination, in particular of the innate immune system. The analysis of 12 cell types may lead to more insight.

Authors' reply: We thank the reviewer for this valuable suggestion. In response, we have applied the EpiDISH tool to estimate the proportions of 12 cell types in our data. We have performed a sensitivity analysis for the suggestively significant CpG sites in the original 6-cell type model ($P < 1e-5$ shown in the supplementary tables), adjusting for the 12-cell types instead. This adjustment includes results for both the BCG effect (changes in DNA methylation over time

after BCG vaccination) and cytokine responses (baseline DNA methylation and cytokine responses). For the BCG effect, 263 CpG sites were tested in total (Comparing T0-T14-T90, T14-T0, T90-T0, and T90-T14). Of these, 214 CpG sites (81.4%) are also suggestively significant ($P < 1 \times 10^{-5}$) in the model adjusting for 12 cell types. These results have been included in the supplemental files (Table S8). For cytokine responses, all the results showed a consistent direction of effect with the original analysis, and 39% of the sites remained suggestive significant ($P < 1 \times 10^{-5}$) in the adjusted model. These results have also been added to Table S17.

We have incorporated these sensitivity results into the manuscript as follows:

Page 23, Line 954-961 of Methods:

“Considering that cell type heterogeneity can be a significant confounder in the EWAS study, we also applied EpiDISH [PMID: 31710662], which predicts the proportions of 12 cell types, to achieve a higher resolution of blood cells composition. Sensitivity analyses were performed on the top CpG sites associated with BCG effect (changes in DNA methylation over time after BCG vaccination, $P < 1 \times 10^{-5}$) and TI (baseline DNA methylation and cytokine productions, including IL6 and IFN- γ , $P < 1 \times 10^{-5}$).”

Page 8, Line 277-280 of Results:

“In the sensitivity analyses incorporated 12 cell types estimated by EpiDISH²⁸, a total of 263 CpG sites were tested ($P < 1 \times 10^{-5}$, including T0-T14-T90, T14-T0, T90-T0, and T90-T14). Among these, 214 CpG sites (81.4%) remained suggestive significant ($P < 1 \times 10^{-5}$) after adjusting for the 12 cell types (Table S8).”

Page 11, Line 455-459 of Results:

“In the sensitivity analyses, where 12 cell types were included as covariates in the model, all the results showed a consistent direction of effect with the original analysis, and 39% of CpG sites remained suggestive significant ($P < 1 \times 10^{-5}$) (Table S17). This indicates that some of the CpG sites might be driven by specific immune cell types that should be interpreted with caution.”

Top lines of Table S8, BCG-associated CpG sites in the model adjusted for 12 cell types

			Adjustment of 12 cell types estimated by EpiDISH	
group	CpG	Pval_Origin	Pval_Adj	P<1e-5 in adj
T0-T14-T90	cg18149745	6,12E-14		5,72E-08 Y
T0-T14-T90	cg06013215	4,83E-12		1,62E-10 Y
T0-T14-T90	cg26592283	7,14E-10		3,31E-09 Y
T0-T14-T90	cg00418999	4,46E-09		1,06E-11 Y
T0-T14-T90	cg13763783	2,09E-08		4,42E-10 Y
T0-T14-T90	cg19346823	5,96E-08		2,27E-07 Y
T0-T14-T90	cg10710218	3,48E-07		4,56E-06 Y
T0-T14-T90	cg16926388	3,72E-07		4,12E-06 Y
T0-T14-T90	cg16548306	4,37E-07		1,78E-08 Y
T0-T14-T90	cg20012885	5,44E-07		5,87E-07 Y

Top lines of Table S17, cytokine production associated CpG sites in the model adjusted for 12 cell types

cytokine	CpG	BETA_origin	SE_origin	P_origin	BETA_Adj	SE_Adj	P_Adj	P<1e-5 in adj
IL6	cg07586956	1,48	0,24	5,40E-10	1,51	0,24	4,10E-10	Y
INFg	cg16685860	0,97	0,16	6,02E-10	0,80	0,19	1,98E-05	N
INFg	cg08236479	1,37	0,24	1,84E-08	1,40	0,24	1,22E-08	Y
INFg	cg16268214	1,16	0,21	3,01E-08	1,11	0,22	5,53E-07	Y
INFg	cg06748688	0,92	0,17	1,04E-07	0,81	0,21	1,43E-04	N
INFg	cg06996609	-1,76	0,34	3,36E-07	-1,69	0,35	1,10E-06	Y
INFg	cg11698354	1,4	0,28	4,31E-07	1,53	0,29	1,04E-07	Y
INFg	cg18591181	-1,26	0,25	4,44E-07	-1,20	0,27	5,69E-06	Y
INFg	cg18079334	0,98	0,19	4,61E-07	0,84	0,21	6,59E-05	N
INFg	cg10404776	1,15	0,23	5,13E-07	1,08	0,25	2,16E-05	N
INFg	cg14346774	-1,62	0,32	5,15E-07	-1,59	0,32	5,63E-07	Y

- 3. A surprising aspect of the manuscript is that replication is sought only after all analyses were done at the very end of the paper. The order should be reversed and replication should be tested immediately after the discovery phase. Given the relatively small replication set, this analysis can focus on consistency rather than formal statistical significance (e.g. direction of effects). The change in order is particularly important because the authors rely on a lenient threshold for suggestive statistical significance.

Authors' reply: We agree with the reviewer for the valuable suggestion. We agree that moving the replication results immediately after the discovery phase would help readers better understand the results. As such, we have reorganized the manuscript and placed the replication results at the end of each section of the discovery analysis.

- 4. The authors now only report scaled DNA methylation differences (e.g. figure 2). While this looks visually convincing, it removes the opportunity to gauge the actual effect sizes and their interpretation. It is important to (also) show the differences as methylation fractions (beta-value

scale; which may for example be set relative changes to those at t=0). This is relevant to know to what extent the DNA methylome is changed and could be of functional relevance.

Authors' reply: we thank the reviewer for this insightful comment. The actual changes in the methylation beta value are between -0.03 to 0.03, as shown in Supplementary Figure S3. This is expected, as vaccination is unlikely to cause dramatic changes in methylation, unlike what is seen in cancer studies [PMID: 39134824]. To better visualize changes in methylation levels, we used scaled values, which allow for a clearer understanding of how methylation levels change after BCG vaccination. In response to the reviewer's suggestion, we have now included the beta-value scale in Figure S3 to provide additional details to the reader.

[PMID: 39134824]: Smith ZD, Hetzel S, Meissner A. DNA methylation in mammalian development and disease. *Nat Rev Genet.* 2025;26(1):7-30. doi:10.1038/s41576-024-00760-8

- 5. The authors should report and, if necessary, address statistical inflation of test statistics.

Authors' reply: We thank the reviewer for this important point. In response, we have reported the statistical inflation of all the EWAS in the supplementary tables (Table S28). Page 22, Line 909: “The statistical inflation of all the EWAS results is provided in Table S28.”

Top Lines of Table S28, Inflation factor (lambda) of each EWAS model in this study

Table S28. Inflation factor (lambda) of each model in this study			
Models	lambda		
1. Models of BCG effect			
all T0-T14-T90	1.34		
all T14-T0	1,11		
all T90-T0	1,15		
all T90-T14	1,36		
2. trained immunity ~ baseline DNAm			
IFN- γ	1,09		
IL1 β	1,08		
IL6	1,04		
TNF α	1,02		

- 6. The discovery analysis includes batch as covariate. Please, specify which batch (sample plate?).

Authors' reply: We thank the reviewer for this comment. The “batch” covariate refers to the sample plate, and we have clarified this in these methods section accordingly.

Page 22, Line 916

*“To assess the DNA methylation changes induced by BCG vaccination, a linear mix-effects model was performed for each CpG site, with time point as the independent variable, subject ID as the random effect, and age, sex, batch (**sample plate**), and estimated cell counts as the covariates.”*

- 7. A very intriguing finding is that the study highlights the role of T cell responses in BCG vaccination and downplays the responses of monocytes. This would have been missed if monocytes were studied instead of whole blood. Did the authors consider analysis like interaction tests to gain some indication in which cell type the DNA methylation difference occurred? Are transcriptomic data available? This could very much strengthen the insight in and evidence for a functional effect of the DNA methylation differences observed and - e.g. using cell type specific expression patterns - could pinpoint the putative cell type involved. Irrespectively, the manuscript will be improved by a discussion of the various alternative explanations. More discussion highlighting the involvement of T cells in contrast to innate immune cells is justified. In contrast, the discussion mentions 'innate immune cells' and 'epigenetic scar'. Given that whole blood is studied and associations were found for T cell responses, this conclusion cannot be founded on the data.

Authors' reply: we thank the reviewer for this insightful comment. In our study, we emphasized “T cell responses” because we found that INF- γ , which represents a T-cell-derived immune response, was associated with 41 CpG sites in the analysis of baseline DNA methylation and ex-vivo cytokine production. In contrast, we did not find any significant associations between baseline DNA methylation and IL-1 β or TNF- α cytokine response, except for one CpG site that showed an association with IL-6 production ($P=5.40 \times 10^{-10}$). We agree that identifying the specific cell types involved in the DNA methylation difference would be valuable for interpreting the data. Although single-cell RNAseq was performed in this cohort [PMID: 37155329], the sample size ($n=38$) was too small to identify significant associations between methylation levels and gene expression. On the other hand, cell count data assessed by flow cytometry are available, and we correlated the INF- γ associated CpG sites with the cell counts of major blood cell types (CD4+ T cells, CD8+ T cells, B cells, NK cells, NK T cells, monocytes, neutrophils, and eosinophils). Our analysis showed that T cells (both CD4+ and CD8+) were associated with the majority of the CpG sites, suggesting that T cell numbers are associated with the INF- γ -associated DNA methylation patterns (Figure S6C). Furthermore, we performed interaction analysis to assess the interaction between DNA methylation and CD4+/CD8+ T cells in association with INF- γ . However, no significant interaction effects were observed. Hereby, we have included the CpG-cell counts association in the Results, and we modified the Discussion regarding “innate immune cells”.

Page 10, Line 355-357

“We associated INF- γ related CpG sites with cell counts measured by flow cytometry and identified that T cell numbers (both CD4+ and CD8+) were associated with the greatest number of CpG sites (Figure S6C).”

Figure S6C. INF- γ associated CpG sites and cell counts measured by flow cytometry

Page 18, Line 762-765

“Previous research has indicated that IFN-γ plays a critical role in amplifying the trained immunity response at the transcriptomic level (13). Our findings may suggest that crosstalk between innate immune cells and adaptive immune cells may be crucial for the development of the trained immunity at the epigenetic level.”

[PMID: 37155329] Li, W., Moorlag, S.J.C.F.M., Koeken, V.A.C.M., Röring, R.J., de Bree, L.C.J., Mourits, V.P., Gupta, M.K., Zhang, B., Fu, J., Zhang, Z., et al. (2023). A single-cell view on host immune transcriptional response to in vivo BCG-induced trained immunity. *Cell Rep* 42, 112487. <https://doi.org/10.1016/j.celrep.2023.112487>.

- 8. The link with aging put forward in the discussion is superficial.

Authors’ reply: we thank the reviewer for the comment and agree the linking of BCG-associated CpG sites with aging is an interesting topic. However, the topic of aging is beyond the main scope of the present study. Therefore, we deleted the “aligning with the concept of epigenetic aging drift²⁸”. On page 8, line 262.

- 9. After the main analysis, a series of follow-up analysis are presented which are not all performed with similar rigor or clear research question.

Authors’ reply: We thank the reviewer for this important observation. In response, we have carefully reviewed our follow-up analyses to ensure clarity in both the research questions and

the rigor of the methods applied. Please find below our detailed replies to each specific concern:

9.1 DNA methylation changes are associated with protein levels. Some associations are found but the functional role and specific link with the outcomes of interest remain unclear. Moreover, despite testing many proteins, associations with p-values that were merely <0.05 are pushed as of interest. As the last sentence speculates on a causal role which is not supported by the type of analyses done nor supported by the outcome.

Authors' reply: We thank the reviewer for this valuable comment. In the protein associations analyses, we reported CpG sites with FDR correction, as stated: "*We found that the DNA methylation changes in four CpG sites identified from the T14 vs T0 comparison were significantly associated with alterations in two proteins upon BCG vaccination, with $FDR < 0.05$ (Table S10).*" However, given the limited number of CpG sites passing FDR correction ($n=4$), it is difficult to explore the overall pattern of 4 CpG sites. To explore an overview of the patterns of how the CpG sites are associated with various proteins, we applied a more lenient nominal significance threshold ($P < 0.05$). This exploratory analysis revealed a consistent pattern, wherein CpG sites exhibiting increased methylation at T14, compared to T0, were inversely correlated with proteins that exhibited a decrease at T14 relative to T0 (Figure S5A). Regarding the last sentence mentioned here, we have adjusted our interpretation to adopt a more cautious tone.:

"This suggests that short-term DNA methylation changes may play a role in the reduction of circulating inflammatory proteins induced by BCG vaccination." (Page 9, Line 381)

9.2 Baseline DNA methylation is correlated to ex vivo cytokine production. This analysis does not link to the research question whether vaccination does induces a DNA methylation response. The number of CpGs associated with outcomes is mentioned, which is not informative. Some analyses are put forward as 'interestingly (line 286) but it remains unclear to the reader why this is interesting because not link with the research question is made. All in all, this analysis seems out of place.

Authors' reply: We thank the reviewer for this insightful comment. The reviewer is correct that this analysis does not directly address the research question of whether vaccination induces a DNA methylation response. However, the purpose of this analysis was to answer whether baseline methylation status is associated with trained immunity response (TI) *response, which was defined as the fold changes of cytokine productions stimulated with S. aureus at T90 compared to T0 after BCG vaccination*". In other words, our goal was to determine if baseline DNA methylation could serve as a "predictor" of the trained immunity response induced by BCG vaccination. To our best understanding, baseline DNA methylation may reflect host factors influenced by prior exposures and interactions with genetic background. Our analysis identified that baseline DNA methylation was associated with IFN- γ response and explained a considerable portion of its variance. This finding suggests that differences in individuals' IFN- γ response to BCG vaccination can, to a certain extent, be attributed to their baseline methylation status. We believe this is an important insight that contributes to understanding the heterogeneity of vaccine-induced responses.

To clarify this rationale, we have added the following sentence at the beginning of the relevant paragraph on Page 9, Lines 387-390:

"Next, to further understand whether differences in baseline DNA methylation influence BCG-induced trained immunity(TI), we assessed the association between baseline DNA methylation status and BCG-induced TI responses. TI responses were defined as the fold changes of cytokine productions stimulated with S. aureus at T90 compared to T0."

9.3 The kisspeptin result is potentially interesting but it is not clear where it comes from. Is it a candidate gene analysis? Why this one and not any of the many other potentially interesting ones? Where 'in this study' was an association with methylation observed?

Authors' reply: We apologize for the confusion. To clarify, we first identified baseline CpG methylation patterns that were associated with IFN- γ production. Next, we performed a pathway enrichment analysis on the genes mapped to these CpG sites. Among the pathways identified, the kisspeptin pathway showed the strongest enrichment, with the lowest P value and the largest number of enriched genes (Figure S6A). To further understand whether kisspeptins modulate IFN- γ production, we performed the functional experiment. Human

peripheral blood mononuclear cells (PBMCs) were incubated for one hour with various concentrations of recombinant kisspeptin-10 (from 1 ng/ml to 100 ng/ml), followed by stimulation with either heat-killed *C. albicans* or *S. aureus* for 7 days. To make this rationale clearer, we added the following sentences at the beginning of the relevant paragraph:

Page 11, Line 465-474:

*“According to the pathway enrichment analysis of IFN- γ associated CpG sites, the kisspeptin receptor system pathway showed the strongest enrichment, with the lowest P value and the largest number of enriched genes (Figure S6A). ... To further understand whether kisspeptins modulate IFN- γ production, we performed a functional validation experiment. Human peripheral blood mononuclear cells (PBMCs) were incubated for one hour with various concentrations of recombinant kisspeptin-10 (from 1 ng/ml to 100 ng/ml), followed by stimulation with either heat-killed *C. albicans* or *S. aureus* for 7 days.”*

9.4 BCG-induced DNA methylation changes and their association with ex vivo cytokine production seem very much on topic. More context on the interpretation of associations and consistency of findings is warranted. Now mainly the number of associations per cytokine are mentioned. This is not particularly informative. Finally, earlier, the authors seem to find it interesting that most differentially methylated CpGs were observed for the T14-T19 comparison, but none of them was associated with ex vivo cytokine production. So, this is worrying?

Authors' reply: We thank the reviewer for raising this important concern. The reviewer asked whether the lack of association between the majority of differentially methylated CpGs for the T90-T14 comparison and ex vivo cytokine production is worrying. We interpret the T90-T14 comparison as representing the recovery phase following BCG vaccination, where epigenetic changes may not be as strongly linked to the trained immunity response. Thus, while the lack of association is unexpected, we do not consider it a major issue. To reflect this interpretation, we have added the following text on pages 12-13 lines 522-531.

“These findings suggest that the BCG-induced long-term DNA methylation changes, rather than those observed during the vaccine phase (T14-T0) or recovery-phase (T90-T14) changes, may play a more important role in the trained immunity response.”

We also agree with the reviewer that the association between BCG-induced DNA methylation changes and *ex-vivo* cytokine production is highly relevant. Beyond simply reporting the number of associations, we have provided biological interpretations of these associations, as detailed below:

Page 12, Line 506-515

“The trained immunity response, as assessed by IL-1 β production in response to a heterologous stimulus from the model T90-T0 was associated with enrichment in changes in DNA methylation for genes involved in Kinesins and Golgi-to-ER retrograde traffic. Kinesins are motor proteins that ensure the transport of cellular cargo, with an emerging role in supporting immune processes as well. On the other hand, changes in IFN- γ heterologous production were enriched in changes in DNA methylation of genes involved in the mTOR signaling pathway (well-known to be involved in T-cell activation and trained immunity), the VEGFA-VEGFR2 signaling pathway, and other immune-related pathways (e.g., IL-2 family signaling and IL-18 signaling pathways). These findings provide a further understanding of the functional implications of the DNAm-C in relation to cytokine production changes.”

Subsequently, we performed mediation analyses to further understand the role of DNA methylation changes in trained immunity, and the CpG sites with significant mediation effects were listed and discussed:

Page 14, Line 576-578

“For instance, cg25926804, located near the TMEM75 and GSDMC gene, mediated 28.6% of the effect of rs6991078 on TI(IFN- γ) (Figure 4D). GSDMC is known to be involved in antimicrobial defense response.”

In terms of consistency of findings, unfortunately, we were unable to replicate these results in the replication cohorts due to the limited sample size of only 17 individuals. This limitation arose because in this association analysis, we first regressed out cell counts and batch effects at each time point from the DNA methylation M values, and subsequent analyses were conducted using the resulting residuals. We have added this as a limitation on page 19 lines 817-818.

“Further validation of the association between BCG vaccination-induced methylation change and trained immunity is still needed”.

9.5 The mediation analysis is of potential interest, but due to inherent limitations cannot be simply interpreted as causal effects (and also not inform on the persistence of methylation differences).

Authors’ reply: we agree with the reviewer that the mediation does not equate to causation and cannot directly inform on the persistence of methylation differences. To address this limitation, we have carefully revised the manuscript to tone down any claims regarding mediation results. Specifically, we have replaced the “causality” with “mediation effect”.

9.6 The sex-specificity analysis is not robust. Most main effects are not formally significant after correcting for multiple testing, this sub-group analysis is not supported by sample/effect sizes and results are at best speculative.

Authors’ reply: we thank the reviewer for raising this important point. We agree that the sex-specificity analysis was performed on sub-groups with smaller sample sizes, which inherently limits the robustness of the results. We did not identify any genome-wide significant CpG sites associated with the effect. Instead, we only identified suggestive significant CpG sites that indicated potential association patterns differing between males and females. Our motivation for conducting the sex-specific analysis stemmed from a previous study by our group, which identified sex-specific inflammatory protein changes induced by BCG vaccination [PMID: 32692728]. This suggests that BCG-induced biological changes may be sex-specific, warranting further investigation into sex-specific methylation changes, an area that remains understudied.

Although part of the female-specific findings was replicated in the validation cohort, we acknowledge that the replication remains underpowered and further validation is needed. Therefore, we also added the following limitation to Discussion:

Page 19, Line 818-821

“Fourth, we did not identify genome-wide significant CpG sites associated with BCG effect in male and female subgroups (FDR<0.05), and thus the sex-specific analyses are restricted by the relatively small sample size and the male-specific results lack external validation.”

[PMID: 32692728]: Koeken VA, de Bree LCJ, Mourits VP, et al. BCG vaccination in humans inhibits systemic inflammation in a sex-dependent manner. J Clin Invest. 2020;130(10):5591-5602. doi:10.1172/JCI133935

- 10. The manuscripts now at times mentions that 'functional validation experiments' were done. Perhaps the ex-vivo cytokine production assays are meant. However, all analyses in the manuscript rely on association. This is not a bad thing, but should be phrased as such and described accordingly in the discussion.

Authors' reply: we apologize for the unclear description of the functional validation experiments. These experiments refer indeed to the ex-vivo cytokine production assays, but mainly to the experiment of kisspeptin. To make it clearer, we rephrase it as **functional validation experiments of kisspeptin** in the manuscript.

Page 18, Line 752-754

*“Indeed, **functional validation experiments of kisspeptin** demonstrate their modulatory effects on IFN- γ production, strengthening the hypothesis of their impact on long-term immune responses.”*

- 11. No non-vaccinated controls (placebo) were included in the study. Could seasonal or other effects have contributed to findings?

Authors' reply: we thank the reviewer for raising this important point. We agree that seasonal effects could potentially confound the results. To evaluate whether the significant CpG sites identified in our study are influenced by the seasonality, we performed two additional analyses:

1) Sensitivity analyses: For the BCG-associated CpG sites, we included seasonality as an additional covariate in the model. The seasonality was calculated based on the description in [PMID: 27814508]. The results show that 71.5% of CpG sites remain suggestive significant ($P < 1e-5$) after adjusting for seasonality (Table S8).

2) Correlation with seasonality: We assessed whether top CpG sites (BCG-associated and cytokine-associated) were correlated with seasonality at baseline. The results show that 95.4% of BCG-associated CpG sites are not significantly associated with seasonality ($P > 0.05$, Table S9), and none of the CpG sites associated with INF- γ and IL-6 production capacity are associated with seasonality ($P > 0.05$, Table S18). These results confirmed that seasonality does not significantly influence the top CpG sites identified in our study. We have added the results from these sensitivity analyses to the manuscript for clarity:

Page 8, Line 281-284

“Additionally, we assessed the impact of seasonality on the BCG-associated CpG sites. The results showed that 71.5% of the top CpG sites remained suggestive significant ($P < 1 \times 10^{-5}$) after seasonality adjustment (Table S8), and 95.4% of these BCG-associated CpG sites are not associated with seasonality ($P > 0.05$, Table S9).

Page 11, Line 460-462

“Additionally, sensitivity analyses revealed that these CpG sites associated with IFN- γ or IL-6 production were not associated with seasonality ($P > 0.05$, Table S18).”

Top Lines of Table S8, BCG-associated CpG sites in the model additionally adjust for seasonality

group	CpG	Pval_Origin	Adjustment of 12 cell types estimated by EpiDISH		Additional adjustment for seasonality	
			Pval_Adj	P<1e-5 in adj	Pval_Adj	P<1e-5 in adj
T0-T14-T90	cg18149745	6,12E-14		5,72E-08 Y		8,16E-13 Y
T0-T14-T90	cg06013215	4,83E-12		1,62E-10 Y		1,45E-09 Y
T0-T14-T90	cg26592283	7,14E-10		3,31E-09 Y		1,49E-11 Y
T0-T14-T90	cg00418999	4,46E-09		1,06E-11 Y		9,37E-11 Y
T0-T14-T90	cg13763783	2,09E-08		4,42E-10 Y		1,99E-08 Y
T0-T14-T90	cg19346823	5,96E-08		2,27E-07 Y		1,67E-06 Y
T0-T14-T90	cg10710218	3,48E-07		4,56E-06 Y		1,58E-07 Y
T0-T14-T90	cg16926388	3,72E-07		4,12E-06 Y		1,84E-07 Y
T0-T14-T90	cg16548306	4,37E-07		1,78E-08 Y		2,28E-07 Y
T0-T14-T90	cg20012885	5,44E-07		5,87E-07 Y		2,12E-06 Y
T0-T14-T90	cg06221058	7,24E-07		8,47E-09 Y		4,18E-07 Y

Top Lines of Table 18, the associations between top CpG sites and seasonality

Table S18. The associations between top CpG sites associated with cytokine production (IL6 and IFN- γ) and seasonality

Cytokine	CpG	pheno	estimate	se	Pval	p.adj
IL6	cg07586956	season	1,81E-02	1,54E-02	0,24	0,99
INFg	cg16685860	season	-3,33E-03	2,84E-02	0,91	0,99
INFg	cg08236479	season	-2,33E-02	1,46E-02	0,11	0,94
INFg	cg16268214	season	8,22E-03	2,07E-02	0,69	0,99
INFg	cg06748688	season	2,41E-03	2,54E-02	0,92	0,99
INFg	cg06996609	season	-3,82E-03	1,35E-02	0,78	0,99
INFg	cg11698354	season	-1,78E-02	1,58E-02	0,26	0,99
INFg	cg18591181	season	1,86E-02	1,77E-02	0,29	0,99
INFg	cg18079334	season	-4,11E-03	2,39E-02	0,86	0,99
INFg	cg10404776	season	-8,74E-03	1,75E-02	0,62	0,99
INFg	cg14346774	season	-1,97E-02	1,41E-02	0,16	0,99

[PMID: 27814508]: Ter Horst R, Jaeger M, Smeekens SP, et al. Host and Environmental Factors Influencing Individual Human Cytokine Responses. *Cell*. 2016;167(4):1111-1124.e13. doi:10.1016/j.cell.2016.10.018

- 12. The authors study the BCG response. It is not clear why the results can be generalized to the concept of trained immunity. This requires more support or toning down. The apparent limited involvement of innate immune cells should be considered. Of note, the second paragraph of the introduction only refers to studies of BCG vaccination and no other exposures.

Authors' reply: we thank the reviewer for this insightful comment. In this study, BCG was used as a model to study trained immunity. Trained immunity response(TI) response, is defined in this study as the fold changes in cytokine production stimulated with the heterologous stimulus *S. aureus* at T90 compared to T0. Specifically, we measured four cytokines IL-1 β , TNF- α , INF- γ and IL-6.

The reviewer is completely correct that while the findings presented here concern specifically BCG-induced trained immunity, the conclusions cannot be automatically extended to the

entire trained immunity field. However, other vaccines, including Influenza and Measles-containing vaccines, have been shown to induce trained immunity in humans in addition to BCG vaccination [PMID: 39278362], and it is likely that DNA methylation plays an important role in the effects of these vaccines as well. We do agree nevertheless that our study is restricted to BCG-induced trained immunity: thus, in order to limit our conclusions to BCG vaccination, we modified the description of trained immunity in the manuscript to “BCG-induced trained immunity”, for example, Page 4, Line 134-136: “*The fold change (FC) of cytokine production between T90 and T0, referred to as trained immunity markers in this study, was used as a measure of **BCG-induced trained immunity response.***” And Page 9 Line 387-390: “*Next, to further understand whether differences in baseline DNA methylation influence BCG-induced trained immunity(TI), we assessed the association between baseline DNA methylation status and BCG-induced TI responses. TI responses were defined as the fold changes of cytokine productions stimulated with S. aureus at T90 compared to T0 (Figure 1).*”

[PMID: 39278362]: Netea MG, Joosten LAB. Trained innate immunity: Concept, nomenclature, and future perspectives. J Allergy Clin Immunol. 2024;154(5):1079-1084.
doi:10.1016/j.jaci.2024.09.005

- 13. The limitations paragraph in the discussion is overly optimistic and should provide a complete overview of limitations and alternative explanations for results should be considered.

Authors' reply: according to the suggestions of the reviewer we added the following sentences to the limitation paragraph:

Page 19-20 Line 811-846

“This study has some limitations that should be acknowledged. First, DNA methylation profiles were measured from whole blood samples. Since DNA methylation measurements are cell-type specific, the use of whole blood samples may mask the cell-type specific methylation changes. Second, the DNA methylation analysis conducted using the EPIC array only covers approximately 3% of the CpG sites in the genome. Third, while we were able to partially replicate our results in an independent cohort, the validation study had limited statistical power due to the limited sample size, and the replication cohort only includes two time points, further validation on the “late” effect is still needed. Fourth, we did not identify genome-wide

significant CpG sites associated with BCG effect in male and female subgroups (FDR<0.05), and thus the sex-specific analyses are restricted by the relatively small sample size and the male-specific results lack external validation. Fifth, we were not able to assess the transcription of the genes influenced by the CpG sites at the expression level, which limits the interpretation of our results. Sixth, this study only assessed the role of BCG-induced DNA methylation changes in ex vivo cytokine production through association and mediation analyses, further functional studies are still needed to explore the molecular mechanism. Seventh, the study population consisted primarily of young adults (<30), the generalizability of our findings to older people and the age-different methylation patterns induced by BCG vaccination still deserves further investigation. Finally, while our study provides evidence for the role of BCG-induced DNA methylation changes in cytokine production, alternative explanations should be considered. The lack of a non-vaccinated control group may limit attributing the observed changes to BCG-induced trained immunity.

.”

- 14. The authors use the term persistence. This term should be removed. The authors show that the changes at day 14 are not persistent and for the changes at day 90 we simply do not know what happens later.

Authors' reply: we thank the reviewer for this suggestion. We have removed the term "persistent" and replaced it with "long-term" to more accurately reflect the findings.

- 15. Page 6, line 198: 'the methylated CpG sites' not immediately clear this refers to an increase in methylation in contrast to demethylation.

Authors' reply: We thank the reviewer for pointing this out. To clarify, we have changed the wording to 'methylated (**the increase in methylation**) CpG sites' to specify that this refers to an increase in methylation as opposed to demethylation.

- 16. The existence of a short-term effect on DNA methylation that seems to be reversed from day 14 to 19 and replaced by a distinct longer-term DNA methylation signature is quite interesting. And also seems new. The only previous example I have seen in the literature from a different context is from Tilburg et al (Epigenomics 2020). Are there other examples?

Authors' reply: we thank the reviewer for this insightful comment. In the context of BCG effect on ex vivo cytokine responses, we are not aware of other studies reporting similar patterns of changes. In a broader context, a PubMed search on longitudinal methylation changes identified an additional example: different short-term and long-term DNA methylation changes during clozapine exposure [PMID: 39333502]. However, we believe that the discussion of this research is beyond the scope of this study.

[PMID: 39333502] Gillespie AL, Walker EM, Hannon E, et al. Longitudinal changes in DNA methylation associated with clozapine use in treatment-resistant schizophrenia from two international cohorts. *Transl Psychiatry*. 2024;14(1):390. Published 2024 Sep 27.
doi:10.1038/s41398-024-03102-8

Reviewer #2:

The study investigates the role of DNA methylation in BCG-induced trained immunity using a substantial cohort of 303 healthy volunteers, providing good statistical power. Data is collected at multiple time points (T0, T14, T90), allowing for the assessment of both short-term and long-term effects. DNA methylation data is integrated with cytokine production and plasma inflammatory protein concentrations. The integration of genetic data and the mediation analysis between genetic variants, DNA methylation, and cytokine responses is a strength.

Authors' reply: We thank the reviewer for the detailed and constructive feedback.

Some of the weaknesses of the study are:

1. The use of whole blood DNA may mask cell-type-specific methylation changes. While the authors estimate cell proportions, direct measurement of cell populations or single-cell analysis would have provided more accurate and insightful data.

Authors' reply: we thank the reviewer for this insightful comment and agree that the cell-type-specific methylation changes would have provided more detailed and insightful findings. While we were unable to measure DNA methylation at the single-cell level, we did estimate cell proportions through flow cytometry and use these data to better understand the relationship between cell counts and significant CpG sites (as discussed in reviewer 1 comment 7). Although these analyses offer limited information on cell-type-specific DNA methylation patterns, we believe they still provide valuable insights. Additional DNA methylation patterns in different cell types are not possible due to a lack of enough biological samples. To address this limitation, we have expanded our discussion on this topic:

Page 19 Line 811-814,

“First, DNA methylation profiles were measured from whole blood samples. Since DNA methylation measurements are cell-type specific, the use of whole blood samples may mask the cell-type specific methylation changes.

2. The study doesn't provide a clear mechanistic explanation for how the observed methylation changes contribute to trained immunity.

Authors' reply: we thank the reviewer for this thoughtful comment and acknowledge that the association analyses presented in this study do not allow us to provide a clear mechanistic explanation for how DNA methylation changes contribute to trained immunity. Our study primarily suggests a potential mediatory role of DNA methylation in ex vivo cytokine production, which could serve as a foundation for further investigation into the underlying mechanisms. To address this limitation, we have revised the manuscript to explicitly emphasize the need for additional functional studies:

Page 19-20, Line 823-840

“This study only assessed the role of BCG-induced DNA methylation changes in ex vivo cytokine production through association and mediation analyses, further functional studies are still needed to explore the molecular mechanism.”

3. The authors mention 'monocyte and heterologous T cells' in the paper's title, which is not supported by the data. Although the correlation of methylation changes with neutrophil subtypes has been shown, no specific analysis of monocytes or T cells is presented. The authors should focus the title on the broader findings related to DNA methylation changes in whole blood following BCG vaccination unless they have additional data specifically on monocytes and T cells.

Authors' reply: we apologize for the confusion. The term “monocyte and heterologous T cells” in the title refers to the ex-vivo cytokine production analyzed in our study, Specifically, monocyte-derived cytokines include tumor necrosis factor-alpha (TNF- α), interleukin-1 β (IL-1 β), and interleukin-6 (IL-6), while T cell-derived cytokines are represented by interferon-gamma (IFN- γ). However, we understand that the data presented in the study do not include specific analyses of monocytes or T cells themselves. To clarify this point, we have added the following text to the revised manuscript: Page 4, Line 117-123:

“Using integrative system biological approaches, we assessed the dynamic changes in DNA methylation following BCG vaccination and their associations with BCG-induced immunological memory, focusing on ex vivo cytokine production changes of four cytokines upon S. aureus stimulation. Specifically, monocyte-derived cytokines include tumor necrosis factor-alpha (TNF- α), interleukin-1 β (IL-1 β), and interleukin-6 (IL-6), while T cell-derived cytokines are represented by interferon-gamma (IFN- γ).”

4. The sex-specific effects observed in methylation changes and cytokine responses are interesting findings that could be highlighted more.

Authors' reply: Following the reviewer's suggestion, we highlight the sex-specific effect in the abstract:

“Sex-specific effects were observed in DNA methylation and cytokine responses, highlighting the importance of considering sex in immune studies.”

And in the summary paragraph of the discussion (Page 16 Line 677-679) we added:

“Our results reveal sex-specific effects of BCG-induced DNA methylation changes and cytokine production, highlighting the importance of considering sex differences in immune response studies.”

Some suggestions for change are:

* 5. Only 11 CpG sites were identified as significantly changed (FDR < 0.05) following BCG vaccination, which is a relatively small number for a genome-wide study. The reason for this could be discussed.

Authors' reply: We thank the reviewer for this comment. In our study, we identified only 11 CpG sites that were significantly changed after BCG vaccination (FDR<0.05). One possible explanation would be the use of whole blood, and the cell-type specific effect might be diluted. This limitation has been addressed in the revised version (see also comment 1). Furthermore, other studies investigating complex diseases or environmental exposures using whole blood have also identified a limited number of significant CpG sites [PMID: 32484729, PMID: 33872926, PMID: 34516295, PMID: 29496485]. For example, in an epigenome-wide meta-analysis of childhood asthma, only 14 CpG sites were identified, suggesting that minor DNA methylation changes in whole blood may be harder to detect due to the heterogeneous cell populations in blood samples.

[PMID: 32484729] Eze IC, Jeong A, Schaffner E, et al. Genome-Wide DNA Methylation in Peripheral Blood and Long-Term Exposure to Source-Specific Transportation Noise and Air Pollution: The SAPALDIA Study. *Environ Health Perspect.* 2020;128(6):67003.
doi:10.1289/EHP6174

[PMID: 33872926] Sordillo JE, Cardenas A, Qi C, et al. Residential PM2.5 exposure and the nasal methylome in children. *Environ Int.* 2021;153:106505. doi:10.1016/j.envint.2021.106505

[PMID: 34516295] Hoang TT, Qi C, Paul KC, et al. Epigenome-Wide DNA Methylation and Pesticide Use in the Agricultural Lung Health Study. *Environ Health Perspect.* 2021;129(9):97008. doi:10.1289/EHP8928

[PMID: 29496485] Xu CJ, Söderhäll C, Bustamante M, et al. DNA methylation in childhood asthma: an epigenome-wide meta-analysis. *Lancet Respir Med.* 2018;6(5):379-388. doi:10.1016/S2213-2600(18)30052-3

* 6. The authors could demonstrate how the observed methylation changes affect gene expression or cellular function, e.g., RT-qPCR for genes associated with the significantly changed CpG sites.

Authors' reply: we thank the reviewer for this valuable suggestion. We agree that gene expression data would provide important insights into the functional implications of methylation changes. However, although single-cell RNAseq was performed in this cohort (PMID: 37155329), the sample size (n=38) was too small to find significant associations between methylation levels and expression levels. We have added this limitation to the manuscript: "*We were not able to assess the gene influenced by the CpG sites at the expression level, which limits the interpretation of the results.*" (Page 19, Line 815-817). On the other hand, in the revised version, we have included more information on cell counts measured by flow cytometry (see also reviewer 1 comment 7), to get more cellular function of the identified CpG sites. Unfortunately, no biological samples are available to perform additional analyses, such as RT-qPCR.

[PMID: 37155329] Li, W., Moorlag, S.J.C.F.M., Koeken, V.A.C.M., Röring, R.J., de Bree, L.C.J., Mourits, V.P., Gupta, M.K., Zhang, B., Fu, J., Zhang, Z., et al. (2023). A single-cell view on host immune transcriptional response to in vivo BCG-induced trained immunity. *Cell Rep* 42, 112487. <https://doi.org/10.1016/j.celrep.2023.112487>.

* 7. The rationale for focusing on neutrophils could be explained.

Authors' reply: we apologize for the unclear description. The rationale for focusing on neutrophils is based on both the observed data and previous findings. First, the methylation levels of the 11 BCG-associated CpG sites were significantly correlated with the estimated neutrophil proportions at all three-time points, with the strongest correlations observed at T90.

Additionally, in our previous study on the same cohort [PMID: 33207187], we identified long-term functional reprogramming of neutrophils induced by BCG vaccination. This reprogramming was characterized by increased expression of activation markers and enhanced antimicrobial function. Building on this, we observed strong correlations between BCG-associated CpGs and neutrophil proportion, prompting us to further investigate specific neutrophil subtypes using flow cytometry data from the same paper [PMID: 33207187]. Our analysis revealed that the most significantly correlated subtypes were CD10+ CD66b, CD10+ CD62L, and PDL1-CD62LP neutrophils. We therefore modified our description in the manuscript as follows:

Page 6, Line 194-201

“At all three time points, strong correlations were observed between the CpG sites and estimated neutrophil proportion, and T90 showing the strongest association....The long-term functional reprogramming of neutrophils has been previously reported [PMID: 33207187], and our findings provide additional evidence of BCG-induced DNA methylation changes associated with neutrophils.”

[PMID: 33207187] Moorlag SJCFM, Rodriguez-Rosales YA, Gillard J, et al. BCG Vaccination Induces Long-Term Functional Reprogramming of Human Neutrophils. Cell Rep. 2020;33(7):108387. doi:10.1016/j.celrep.2020.108387

* 8. The manuscript reports a mean age of 25 years with a range of 18 to 71. The standard deviation for age should be mentioned. Age is a known confounder in DNA methylation studies. Methylation patterns change with age, and these changes can be substantial. Without knowing the age distribution, it is difficult to determine if the observed methylation changes are truly due to BCG vaccination or if age differences influence them. If there's a wide age range, the effects of BCG vaccination on DNA methylation could differ between younger and older participants. Detailed age statistics, age-stratified analyses, and potentially exploring the age-vaccination interaction effects on DNA methylation changes could be explored.

Authors' reply: we thank the reviewer for highlighting the importance of age-related analyses in DNA methylation studies. We have added additional age-related statistics and analyses to the manuscript as follows:

*“The mean age of participants was 25 years (**standard deviation 10.55**, ranging from 18 to 71), **with 86.3% of participants under 30 years old (Figure S1)**”* (Page 4, Line 140-141). Given that most participants were young adults, we performed sensitivity analyses on the BCG-associated CpG sites in a subgroup of participants with age >30 (n=39). The results have been added to the supplementary tables (Table S8). Among this subgroup, 21.3% CpG sites associated with the BCG effect remained significant after FDR correction (FDR<0.05), and 46% reached the nominal significance (P<0.05). These findings suggest that some CpG sites might be influenced by the age difference, underscoring the need for further studies in an older population. We therefore added the following sentences to our manuscript:

Page 8, Line 284-289

“Given that the study population primarily consisted of young adults under 30 years old, we further performed sensitivity analyses in the subgroup of participants older than 30. In this subgroup, 46% CpG sites reached nominal significance in the association with BCG effect (Table S8). These findings suggest that the results of this study are most applicable to young adults. Further research in older populations is warranted to better understand these effects.”

Page 20, Line 841-843

“the study population consisted primarily of young adults (<30), the generalizability of our findings to older people and the age-different methylation patterns induced by BCG vaccination still deserves further investigation.”

Top lines of Table S8, BCG associated CpG sites in the subgroup with age>30

Table S8. Sensitivity analyses of BCG-associated CpG sites

			Analysis in subgroup of age>30 (use the same model as original model)		
group	CpG	Pval_Origin	Pval (Age>30)	P.adj (Age>30)	P.adj < 0.05 (Age>30)
T0-T14-T90	cg18149745	6,12E-14	4,99E-02	1,17E-01	N
T0-T14-T90	cg06013215	4,83E-12	1,23E-04	3,99E-03	Y
T0-T14-T90	cg26592283	7,14E-10	1,55E-04	4,31E-03	Y
T0-T14-T90	cg00418999	4,46E-09	1,83E-02	6,21E-02	N
T0-T14-T90	cg13763783	2,09E-08	1,69E-01	2,67E-01	N
T0-T14-T90	cg19346823	5,96E-08	1,64E-02	5,77E-02	N
T0-T14-T90	cg10710218	3,48E-07	1,11E-01	1,99E-01	N
T0-T14-T90	cg16926388	3,72E-07	8,00E-01	8,43E-01	N
T0-T14-T90	cg16548306	4,37E-07	2,74E-03	2,04E-02	Y
T0-T14-T90	cg20012885	5,44E-07	3,18E-02	8,68E-02	N
T0-T14-T90	cg06221058	7,24E-07	2,58E-04	6,26E-03	Y

* 9. Sample handling details could be mentioned whether the blood samples were immediately processed to isolate DNA or was this done from frozen blood samples.

Authors' reply: We thank the reviewer for this suggestion. To clarify, DNA methylation measurements were performed using DNA extracted from frozen blood samples. For the ex-vivo cytokine production measurement, fresh blood samples were used. We have added this information to the methods section as follows.

Page 21, Line 873: *"DNA was isolated from frozen whole blood samples..."*

Page 22, Line 893: *"Peripheral blood mononuclear cells (PBMCs) were isolated from EDTA whole blood with Ficoll-Paque (GE Healthcare) density gradient separation, immediately after collection."*

* 10. A control group would be particularly useful for comparing baseline methylation values. This could help identify any pre-existing differences between the vaccinated and unvaccinated groups that might influence the results. A control group would further strengthen causal inferences about the effects of BCG vaccination. If the study period spans different seasons or other environmental changes, a control group could help account for these factors.

Authors' reply: we thank the reviewer for highlighting this critical issue. We agree that including a control group is ideal for a longitudinal study, especially when the study design aims to infer causality or account for external factors. However, in longitudinal studies focused on within-subject changes, as in our case, robust statistical adjustments and sensitivity analyses can usually address many of the concerns that a control group would typically handle.

To assess whether the significant CpG sites identified are influenced by the season variation, we performed two additional analyses (see also reviewer1 comment 11):

1) Sensitivity analyses: For the BCG-associated CpG sites, we included seasonality as an additional covariate in the model. The seasonality was calculated based on the description in [PMID: 27814508]. The results show that 71.5% of CpG sites remain suggestive significant ($P < 1e-5$) after adjusting for seasonality (Table S8).

2) Correlation with seasonality: We assessed whether top CpG sites (BCG-associated and cytokine-associated) were correlated with seasonality at baseline. The results show that 95.4% of BCG-associated CpG sites are not significantly associated with seasonality ($P > 0.05$, Table S9), and none of the CpG sites associated with INF- γ and IL-6 are associated with seasonality ($P > 0.05$, Table S18). These results confirmed that seasonality does not significantly influence the top CpG sites identified in our study. We have added the results from these sensitivity analyses to the manuscript for clarity:

Page 8, Line 281-284

“we assessed the impact of seasonality on the BCG-associated CpG sites. The results showed that 71.5% of the top CpG sites remained suggestive significant ($P < 1 \times 10^{-5}$) after seasonality adjustment (Table S8), and 95.4% of these BCG-associated CpG sites are not associated with seasonality ($P > 0.05$, Table S9).

Page 11, Line 460-462

“Additionally, sensitivity analyses revealed that these CpG sites associated with INF- γ or IL-6 production were not associated with seasonality ($P > 0.05$, Table S18).”

We have also included the following limitation in the discussion on Page 20, line 845-846:

“The lack of a non-vaccinated control group may limit the possibility to attribute the observed changes to BCG-induced trained immunity.”

[PMID: 27814508]: Ter Horst R, Jaeger M, Smeekens SP, et al. Host and Environmental Factors Influencing Individual Human Cytokine Responses. *Cell*. 2016;167(4):1111-1124.e13.
doi:10.1016/j.cell.2016.10.018

Second round of review

Reviewer 1

I appreciate the authors' thoughtful and insightful response. The manuscript is a pleasure to read.

Reviewer 2

My questions were satisfactorily answered by the authors and the necessary changes were made to the manuscript. However, the The authors should consider revising the title.

The manuscript robustly demonstrates T cell-mediated responses, as evidenced by the strong associations with IFN- γ production. The emphasis of the title should be on the T cells, the emphasis on monocytes in the title is not quite justified.